# One Sample is Enough to Make Conformal Prediction Robust

**Soroush H. Zargarbashi**[1]    **Mohammad Sadegh Akhondzadeh**[2]    **Aleksandar Bojchevski**[2]

[1] CISPA Helmholtz Center for Information Security, [2] University of Cologne

`[zargarbashi, akhondzadeh, bojchevski]@cs.uni-koeln.de`

## Abstract

For any black-box model, conformal prediction (CP) returns prediction *sets* guaranteed to include the true label with high adjustable probability. Robust CP (RCP) extends the guarantee to the worst case noise up to a pre-defined magnitude. For RCP, a well-established approach is to use randomized smoothing since it is applicable to any black-box model and provides smaller sets compared to deterministic methods. However, smoothing-based robustness requires many model forward passes per each input which is computationally expensive. We show that conformal prediction attains some robustness even with *a single forward pass on a randomly perturbed input*. Using any binary certificate we propose a single sample robust CP (RCP1). Our approach returns robust sets with smaller average set size compared to SOTA methods which use many (e.g. $\sim 100$) passes per input. Our key insight is to certify the conformal procedure itself rather than individual conformity scores. Our approach is agnostic to the task (classification and regression). We further extend our approach to smoothing-based robust conformal risk control.

## 1  Introduction

Modern neural networks return uncalibrated probability estimates [14], and other uncertainty quantification methods (like Bayesian and ensemble models, Monte-Carlo dropout) are computationally expensive. Additionally, these methods do not usually provide formal statistical guarantees. Instead, conformal prediction (CP) is a post-processing method returning prediction *sets* with a distribution-free and model-agnostic coverage guarantee, ensuring that the true answer is in the set with an adjustable high probability. To apply CP, we need a conformity[1] score function $s(\boldsymbol{x}, y)$ capturing the agreement between $\boldsymbol{x}$, and $y$ (e.g. softmax). We compute a conformal threshold over a holdout set of calibration points, and for the test points, we form the set as all labels with scores exceeding that threshold. These sets are guaranteed to include (cover) the true label with $1 - \alpha$ probability [2].

As shown in Fig. 1-left (the red dashed line), this guarantee breaks by an unnoticeably small natural or adversarial noise to the test points – the empirical coverage drastically decreases by an imperceptible perturbation. Note that from this point forward, we call adversarial or natural noises as "perturbations", not to be confused with the noise we (as the defender) introduce on purpose. Robust CP (RCP) extends this guarantee to worst case bounded perturbations, ensuring that the perturbed input $\tilde{\boldsymbol{x}}$ is covered with the same or higher probability as the clean $\boldsymbol{x}$, if $\boldsymbol{x}$ is perturbed up to a known magnitude $r$ (e.g. $\|\tilde{\boldsymbol{x}} - \boldsymbol{x}\|_2 \leq r$). Previous RCP approaches find the highest/lowest possible conformity score within the perturbation ball, and replace the score with the worst-case bound[12, 16, 26, 27]. These bounds can be computed either analytically (Lipschitz bound or verifiers) or through randomized smoothing. Here, the trade-off is between the computational cost and the guaranteed robust radius – analytical methods are robust to very smaller magnitudes of perturbation, while they only require a single forward pass

---

[1] Many works define CP via a non-conformity (disagreement) score. The setups are equivalent with a change in the score's sign. Our robustness results are invariant to this definition.

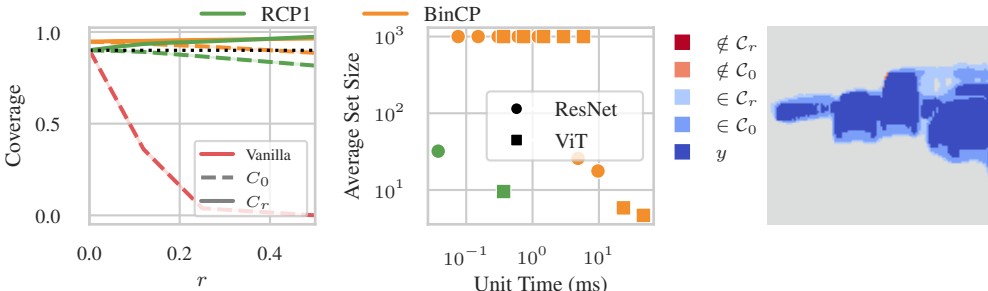

Figure 1: [Left] Coverage of vanilla CP, our robust RCP1, and the SOTA BinCP under adversarial attack (§ B). $\mathcal{C}_r$ denotes sets with robustness guarantee up to radius $r$, and $C_0$ is guaranteed only for clean points – but still using same process only with $r = 0$. Even sets from $\mathcal{C}_0$ show significantly higher coverage compared to vanilla due to randomization. [Middle] The average time to compute, and the average set size for both RCP1 and BinCP, with ResNet and ViT models on the ImageNet dataset; both axis are log-scaled and pareto-optimal points are at the lower-left. RCP1 is more efficient. Both plots are with $\sigma = 0.5$. [Right] Smoothing-based robust conformal risk control. We show the coverage and miscoverage of the RCP1 mask for the class "car" in the segmentation task. Here risk is set to false negative rate.

per input. In contrast, while randomized smoothing needs many model forwards per single input, it provides robustness to significantly larger radii, and it applies over any black-box model.

Smoothing is to augment the input with random noise and inference from the distribution of the model's output over smooth inputs instead. For example, consider adding isotropic Gaussian noise $\epsilon \sim \mathcal{N}(0, \sigma^2 I)$ to the input $x$, and defining the smooth score as the mean of the distribution $\mathbb{E}_\epsilon[s(x + \epsilon, y)]$. Regardless of the original model, the smooth model changes slowly near $x$, as the two distribution $x + \epsilon$, and $\tilde{x} + \epsilon$ have a large overlap. This leads to model-agnostic upper bounds on the score, around $x$. The upper bound is then used to decide whether a label is added to the prediction set. An important drawback here is the computational overhead. The score function (e.g. the expectation of the smooth scores) must be estimated via Monte-Carlo sampling. By reducing the number of samples, the confidence intervals for the mean widen, and the size of the prediction sets quickly increases.

We answer this question: *Can we design smoothing-based RCP without introducing computational overhead?* Interestingly, we show the vanilla CP combined with noise-augmented inference already has robust behavior (green dashed line in Fig. 1-left). With that, we define RCP1 (robust conformal prediction with one sample) that returns robust sets using a single inference per point. In practice, the resulting sets have the same guarantee, and similar size, to the previous SOTA which needs around 100 samples per input instead. By nullifying the need for sampling, we can use even larger models (like vision transformers) to return even smaller sets (see Fig. 1-middle). Importantly, we do not compete with the SOTA equipped with unlimited sampling (compute) budget. Instead, we propose a compute-friendly alternative that still produces small prediction sets in regimes infeasible for the other smoothing-based RCP methods (large models and limited computational power).

RCP1 is similar to vanilla CP only with two changes: (i) we use noise-augmented input (from the smoothing) to compute the scores, and (ii) we calibrate with a conservative $1 - \alpha'$ nominal coverage chosen such that our certified lower bound (in § 3) remains above $1 - \alpha$. RCP1 works with any binary certificate (see § 3.1), is agnostic to the model, the distribution of inputs, and score function, and interestingly, it is task independent – same binary certificate works for both classification, or regression. We use a similar process to define a smoothing-based conformal risk control (Fig. 1-right).

## 2  Background

CP requires a holdout set of labeled calibration points $\mathcal{D}_n = \{x_i, y_i\}_{i=1}^n$ that are exchangeable with the future test point $x_{n+1}$. From the model's output, we define a score function $s : \mathcal{X} \times \mathcal{Y} \to \mathbb{R}$ where it quantifies the agreement between $x$, and $y$, e.g. softmax; see § C for other scores. Vovk et al.

[22] show that under exchangeability (vanilla setup) the set $\mathcal{C}_0(\boldsymbol{x}_{n+1}) = \{y : s(\boldsymbol{x}_{n+1}, y) \geq q\}$ for $q = \mathbb{Q}\left(\alpha; \{s(\boldsymbol{x}_i, y_i) : (\boldsymbol{x}_i, y_i) \in \mathcal{D}_n\}\right)$ contains the true label $y_{n+1}$ at least with $1 - \alpha$ probability.

$$\Pr_{\mathcal{D}_{n+1}}[y_{n+1} \in \mathcal{C}_0(\boldsymbol{x}_{n+1})] = \Pr_{\mathcal{D}_{n+1}}[s(\boldsymbol{x}_{n+1}, y_{n+1}) \geq q] \geq 1 - \alpha \tag{1}$$

Here $\mathcal{D}_{n+1} = \mathcal{D}_n \cup \{(\boldsymbol{x}_{n+1}, y_{n+1})\}$, and $\mathbb{Q}(\alpha; \mathcal{A})$ is the $\lfloor \alpha \cdot (1 - \frac{1}{n}) \rfloor$ quantile of the set $\mathcal{A}$. While the coverage guarantee is agnostic to the model (and the score), better model or score functions reflects in properties like the prediction set size ( a.k.a efficiency). While methods like [27] require bounded score, our results are also agnostic to the choice of the score function (bounded or unbounded).

**Threat model.** We consider the worst case (or adversarial) perturbation, which yields a more powerful guarantee compared to probabilistic robustness e.g. from Ghosh et al. [13]. In our threat model, the adversary aims to decrease the empirical coverage below the guaranteed $1 - \alpha$ by adding an imperceptible noise to the test points (evasion). The set of all possible perturbations is defined as a ball $\mathcal{B} : \mathcal{X} \to 2^{\mathcal{X}}$ around the clean input. We define an inverted ball $\mathcal{B}^{-1}$ as the smallest set that contains the original (clean) point from any possible perturbation; i.e. $\forall \tilde{\boldsymbol{x}} \in \mathcal{B}(\boldsymbol{x}) \Rightarrow \boldsymbol{x} \in \mathcal{B}^{-1}(\tilde{\boldsymbol{x}})$. For images, a common threat model is $\ell_2$-norm: $\mathcal{B}_r(\boldsymbol{x}) = \{\tilde{\boldsymbol{x}} : \|\tilde{\boldsymbol{x}} - \boldsymbol{x}\|_2 \leq r\}$ where $r$ is the radius of the perturbation. For symmetric balls like $\ell_2$ we have $\mathcal{B} = \mathcal{B}^{-1}$, but this does not hold in general (e.g. [5]).

**Robust Conformal Prediction (RCP).** Robust CP extends the guarantee in Eq. 1 to the worst case noise. For $\mathcal{B}$, prior works define a robust (conservative) prediction set $\mathcal{C}_{\mathcal{B}}$ satisfying the following

$$\Pr_{\mathcal{D}_{n+1}}[y_{n+1} \in \mathcal{C}_{\mathcal{B}}(\tilde{\boldsymbol{x}}_{n+1}), \forall \tilde{\boldsymbol{x}}_{n+1} \in \mathcal{B}(\boldsymbol{x}_{n+1})] \geq 1 - \alpha \tag{2}$$

Eq. 2 is only meaningful for deterministic sets. We discuss this subtle point in § 3 and § D.1. Earlier smoothing-based RCP methods implicitly remove all inherent randomness, which makes the definition applicable to them. These methods can be summarized with the following two arguments: (i) for the exchangeable $\boldsymbol{x}_{n+1}$ CP covers the true label with $1 - \alpha$ probability, (ii) if the clean $\boldsymbol{x}_{n+1}$ was originally covered by (vanilla) CP, robust CP also covers $\tilde{\boldsymbol{x}}_{n+1}$, because if a clean score is above $q$ its upper bound (over any perturbed input) is also above $q$ [26]. Thus, the vanilla set for $\boldsymbol{x}_{n+1}$ is a subset of the robust set for $\tilde{\boldsymbol{x}}_{n+1}$. This results in robust coverage of *at least* $1 - \alpha$. To account for the inherent randomness in CP, in § 3, we redefine the threat model, and replace the argument (ii) by the following: "the perturbed $\tilde{\boldsymbol{x}}_{n+1}$ has a higher probability to be in the robust prediction set compared to $\boldsymbol{x}_{n+1}$ being in the vanilla set". The new formulation still addresses the worst-case perturbation.

**Certified bounds.** For any function $f$ and ball $\mathcal{B}(\boldsymbol{x})$ define the certified lower bound as $c^{\downarrow}[f, \boldsymbol{x}, \mathcal{B}] \leq \inf\{f(\boldsymbol{z}) : \boldsymbol{z} \in \mathcal{B}(\boldsymbol{x})\}$, and similarly $c^{\uparrow}[\cdot, \cdot, \cdot]$ as the upper bound (with sup). With this definition, for each $\boldsymbol{x}$ we have $\forall \tilde{\boldsymbol{x}} \in \mathcal{B}(\boldsymbol{x}), c^{\downarrow}[s(\cdot, y), \boldsymbol{x}; \mathcal{B}] \leq s(\tilde{\boldsymbol{x}}, y) \leq c^{\uparrow}[s(\cdot, y), \boldsymbol{x}, \mathcal{B}]$ where we plug in the score function for $f$. Zargarbashi et al. [27] show that given these certified bounds within $\mathcal{B}$, the conservative sets defined either as $\mathcal{C}_{\mathcal{B}}(\boldsymbol{x}_{n+1}) = \{y : c^{\uparrow}[s(\cdot, y_{n+1}), \boldsymbol{x}_{n+1}; \mathcal{B}^{-1}] \geq q\}$ (test-time RCP), or similarly, $\mathcal{C}_{\mathcal{B}}(\boldsymbol{x}_{n+1}) = \{y : s(\boldsymbol{x}_{n+1}, y) \geq \bar{q}\}$ for $\bar{q} = \mathbb{Q}\left(\alpha; \{c^{\downarrow}[s(\cdot, y_i), \boldsymbol{x}_i; \mathcal{B}] : (\boldsymbol{x}_i, y_i) \in \mathcal{D}_n\}\right)$ (calibration-time RCP) attain $1 - \alpha$ robust coverage.

**Randomized smoothing.** One approach to compute these upper/lower bounds for any black-box model, or score is randomized smoothing. A smoothing scheme $\xi : \mathcal{X} \to \mathcal{X}$ adds a random noise to the input – maps it to a random point close to it. A common smoothing for continuous data (e.g. images) is the Gaussian smoothing $\xi(\boldsymbol{x}) = \boldsymbol{x} + \boldsymbol{\epsilon}$ where $\boldsymbol{\epsilon}$ is an isotropic Gaussian noise $\boldsymbol{\epsilon} \sim \mathcal{N}(\boldsymbol{0}, \sigma^2 \boldsymbol{I})$. While our method works for any smoothing, for easier notation we further use $\boldsymbol{x} + \boldsymbol{\epsilon}$ instead of $\xi(\boldsymbol{x})$.

For any score function $s$, the distribution of the smooth scores $s(\boldsymbol{x} + \boldsymbol{\epsilon}, y)$ changes slowly. This enables us to compute tight bounds on the smooth statistics (mean, quantile, etc.) within $\mathcal{B}$, or $\mathcal{B}^{-1}$. RSCP [12, 24], and CAS [27] set the score function directly to the mean of the distribution. BinCP [26], uses the $p$-quantile instead. These statistics are often intractable to compute and therefore estimated using Monte-Carlo sampling, followed by a finite sample correction. RCP1 however nullifies the need to estimate these statistics. We discuss the related work further in § A.

## 3  RCP1: Robust CP with One Sample

**High level view.** We prove that when the scores are computed on noise-augmented inputs, i.e. using $s(\boldsymbol{x} + \boldsymbol{\epsilon}, y)$ instead of $s(\boldsymbol{x}, y)$ for both calibration and prediction (see Alg. 1), vanilla CP already yields robust prediction sets - and its coverage under perturbation can be bounded. We provide a sketch of

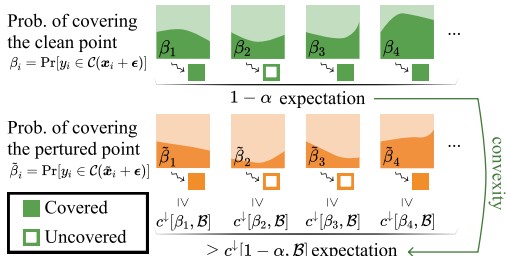

Prob. of covering the clean point
$\beta_i = \Pr[y_i \in \mathcal{C}(\boldsymbol{x}_i + \boldsymbol{\epsilon})]$

$\beta_1$ $\beta_2$ $\beta_3$ $\beta_4$ ...

$1 - \alpha$ expectation

Prob. of covering the perturbed point
$\tilde{\beta}_i = \Pr[y_i \in \mathcal{C}(\tilde{\boldsymbol{x}}_i + \boldsymbol{\epsilon})]$

$\tilde{\beta}_1$ $\tilde{\beta}_2$ $\tilde{\beta}_3$ $\tilde{\beta}_4$ ...

convexity

■ Covered
□ Uncovered

IV $c^\downarrow[\beta_1, \mathcal{B}]$   IV $c^\downarrow[\beta_2, \mathcal{B}]$   IV $c^\downarrow[\beta_3, \mathcal{B}]$   IV $c^\downarrow[\beta_4, \mathcal{B}]$

$\geq c^\downarrow[1 - \alpha, \mathcal{B}]$ expectation

Figure 2: Illustration of the theory behind RCP1. The probabilities $\beta_i$ never need to be computed, since due to the convexity of $c^\downarrow$ we can directly work with $1 - \alpha$. Further description in § 3.

**Algorithm 1.** RCP1; the colored part shows the difference with vanilla CP.

---

**Require:** Calibration set $\mathcal{D}_n = \{(\boldsymbol{x}_i, y_i)\}_{i=1}^n$; nominal coverage $1 - \alpha \in (0, 1)$; score $s : \mathcal{X} \times \mathcal{Y} \to \mathbb{R}$; potentially perturbed test point $\tilde{\boldsymbol{x}}_{n+1}$
1: Compute $s_i \leftarrow s(\boldsymbol{x}_i + \boldsymbol{\epsilon}, y_i) : (\boldsymbol{x}_i, y_i) \in \mathcal{D}$.
2: Set $1 - \alpha' \leftarrow c^\uparrow[1 - \alpha, \mathcal{B}^{-1}]$   ▷ e.g., Gaussian smoothing with $\mathcal{B}_r$: $\Phi_\sigma(\Phi_\sigma^{-1}(1 - \alpha) + r)$.
3: Set $\bar{q}_\alpha = \mathbb{Q}(\alpha', \{s_i\}_{i=1}^n)$.
4: For input $\tilde{\boldsymbol{x}}_{n+1}$ **return**
$$\mathcal{C}_r(\tilde{\boldsymbol{x}}_{n+1}) = \{y : s(\tilde{\boldsymbol{x}}_{n+1} + \boldsymbol{\epsilon}, y) \geq \bar{q}_\alpha\}$$

---

our arguments. To prove it we first assume an abstract value $\beta_{n+1}$ as the coverage probability of a specific clean point $\boldsymbol{x}_{n+1}$, which is over the random noise and the inherent randomness in the score. With $\tilde{\beta}_{n+1}$ as the coverage probability for $\tilde{\boldsymbol{x}}_{n+1}$ (again over augmented input) we show that $\tilde{\beta}_{n+1}$ can be lower bounded using the randomized smoothing certificates. By showing the convexity of the certificate w.r.t. $\beta_{n+1}$, we can directly lower bound the expected coverage over all inputs which is $1 - \alpha$. We never need to compute the abstract $\beta_{n+1}$ or $\tilde{\beta}_{n+1}$. This sketch is illustrated in Fig. 2.

**Limitation of Eq. 2.** The universal quantifier in Eq. 2 implies that for $\geq 1 - \alpha$ fraction of test points, *all* $\tilde{\boldsymbol{x}}_{n+1}$ must be deterministically covered, including the clean $\boldsymbol{x}_{n+1}$. However, many CP methods (like APS [2]) incorporate internal stochasticity (e.g. to break ties), making the coverage event a random variable rather than a binary indicator. This is true even before we add our noise on top. Hence, Eq. 2 does not reduce to Eq. 1 for $r = 0$. For random sets, adversary can reduce the coverage *probability* for each $\boldsymbol{x}_{n+1}$. With $u$ encoding all the (inherent) randomness in the sets, $\mathcal{C}_0$ as the vanilla set, and $\mathcal{C}_r$ as the robust set for a ball $\mathcal{B}_r$, we rewrite the guarantee as:

$$\Pr_{\mathcal{D}_{n+1}}\left[\min_{\tilde{\boldsymbol{x}}_{n+1} \in \mathcal{B}_r(\boldsymbol{x}_{n+1})} \Pr_u[y_{n+1} \in \mathcal{C}_r(\tilde{\boldsymbol{x}}_{n+1}; u)]\right] \geq \Pr_{\mathcal{D}_{n+1}, u}[y_{n+1} \in \mathcal{C}_0(\boldsymbol{x}_{n+1}; u)] \geq 1 - \alpha \quad (3)$$

**General worst-case guarantee.** We prove the lower bound coverage guarantee in RCP1, as implemented in Alg. 1. We state the result in terms of random variables, which have a specific realization in practice. Let $Z_i : i \in [n+1]$ be $n+1$ exchangeable random variables, where $Z_i = (X_i, Y_i)$ for $X_i \in \mathcal{X}$ (e.g. $\mathcal{X} = \mathbb{R}^d$), and $Y_i \in \mathcal{Y}$ (e.g. $\mathcal{Y} = [K]$). Let $s : \mathcal{X} \times \mathcal{Y} \to \mathbb{R}$ be any measurable score function. Let $E_i : i \in [n+1]$ be i.i.d. random variables from a distribution supported on $\mathcal{X}$ (e.g. $E_i \sim \mathcal{N}(\boldsymbol{0}, \sigma^2 \boldsymbol{I})$). We define $\hat{S}_i = s(X_i, Y_i)$, and $S_i = s(X_i + E_i, Y_i)$. Let $\delta \in \mathcal{B}_r$ be any arbitrary perturbation up to radius $r$, define $\tilde{X}_{n+1} = X_{n+1} + \delta$, and $\tilde{S}_{n+1} = s(\tilde{X}_{n+1} + E_{n+1}, Y_{n+1})$ accordingly.

**Proposition 1.** *Let* $q = \mathbb{Q}(\alpha; s(X_i + E_i, Y_i) : (X_i, Y_i) \in \mathcal{D}_n)$. *Given a certified lower bound* $c^\downarrow[\cdot, \mathcal{B}]$ *as later defined in Eq. 4, and* $\mathcal{E}_{n+1} = \{E_i\}_{i=1}^{n+1}$, *for any perturbation* $\delta \in \mathcal{B}_r$ *we have*

$$\Pr_{\mathcal{D}_{n+1}, \mathcal{E}_{n+1}}[s(\tilde{X}_{n+1} + E_{n+1}, Y_{n+1}) \geq q] \geq c^\downarrow[1 - \alpha, \mathcal{B}]$$

*Proof.* Adding i.i.d. noise $E_i$ is permutation equivariant, thus using $S_i$'s doesn't break exchangeability [2]. From Vovk et al. [22] we have $\Pr[S_{n+1} \geq q] \geq 1 - \alpha$ for $q = \mathbb{Q}(\alpha; \{S_i\}_{i=1}^n)$, where the probability is over $\mathcal{D}_{n+1}$, and $\mathcal{E}_{n+1}$. We can rewrite this as

$$\Pr_{\mathcal{D}_{n+1}, \mathcal{E}_{n+1}}[S_{n+1} \geq q] = \mathbb{E}_{\mathcal{D}_{n+1}, \mathcal{E}_n}\left[\Pr_{E_{n+1}}[S_{n+1} \geq q] \mid \mathcal{D}_{n+1}, \mathcal{E}_n\right] = \mathbb{E}_{\mathcal{D}_{n+1}, \mathcal{E}_n}[\beta_{n+1}] \geq 1 - \alpha$$

where we define $\beta_{n+1} := \Pr_{E_{n+1}}[S_{n+1} \geq q \mid \mathcal{D}_{n+1}, \mathcal{E}_n]$. Here, $\beta_{n+1}$ is only a probability over the last noise $E_{n+1}$ (and any other internal randomness in the score) for a fixed $\mathcal{D}_{n+1}$ and $\mathcal{E}_n$. We call $\beta_{n+1}$ the clean instance-wise coverage. Similarly for $\tilde{X}_{n+1}$ we define $\tilde{\beta}_{n+1}$.

We can bound any smooth binary function with an existing certified lower bound $c^\downarrow[\cdot, \mathcal{B}]$. Formally,

$$\Pr_{E_{n+1}}[S_{n+1} \geq q \mid \mathcal{D}_{n+1}, \mathcal{E}_n] = \beta_{n+1} \quad \Rightarrow \quad \Pr_{E_{n+1}}[\tilde{S}_{n+1} \geq q \mid \mathcal{D}_{n+1}, \mathcal{E}_n] \geq c^\downarrow[\beta_{n+1}, \mathcal{B}]$$

Note that here both $\beta_{n+1}$ and $\tilde{\beta}_{n+1}$ share the same $\mathcal{D}_{n+1}$, and $\mathcal{E}_n$ and both are over the random variable $E_{n+1}$ and the inherent randomness of the score. Later in Lemma 1, we show that the function $\mathrm{c}^{\downarrow}[\beta, \mathcal{B}]$ is convex and increasing in $\beta$. This helps us to bound the adversarial coverage guarantee as

$$\Pr[Y_{n+1} \in \mathcal{C}(\tilde{X}_{n+1})] = \Pr[\tilde{S}_{n+1} \geq q] = \mathbb{E}_{\mathcal{D}_{n+1}, \mathcal{E}_n}[\Pr_{E_{n+1}}[\tilde{\beta}_{n+1}]]$$

$$\text{(from certificate)} \geq \mathbb{E}_{\mathcal{D}_{n+1}, \mathcal{E}_n}[\mathrm{c}^{\downarrow}[\beta_{n+1}, \mathcal{B}]]$$

$$\text{(from convexity (Lemma 1), } \mathbb{E}[\mathrm{c}^{\downarrow}[\beta, \cdot]] \geq \mathrm{c}^{\downarrow}[\mathbb{E}[\beta], \cdot]]) \geq \mathrm{c}^{\downarrow}[\mathbb{E}_{\mathcal{D}_{n+1}, \mathcal{E}_n}[\beta_{n+1}], \mathcal{B}] \geq \mathrm{c}^{\downarrow}[1 - \alpha, \mathcal{B}]$$

where the last inequity holds due to vanilla CP and monotonicity. $\qquad\square$

Just like with any other CP with any kind of randomness in the score (e.g. APS), the guarantee only holds marginally over $\mathcal{E}_{n+1}$ and the internal randomness. In other words, the coverage probability is higher than $\mathrm{c}^{\downarrow}[\beta_{n+1}, \mathcal{B}]$ for specific $\tilde{X}_{n+1}$, and $\mathrm{c}^{\downarrow}[1 - \alpha, \mathcal{B}]$ on average, if we draw a random $\mathcal{E}_{n+1}$. If we instead fixed $\mathcal{E}_{n+1}$ and the adversary knows the noise, the guarantee can easily break. Note that $\beta_{n+1}$ is an abstract quantity, the probability that $X_{n+1} + E_{n+1}$ is covered. In principle we can not estimate $\beta_{n+1}$, since the label is not known. Nonetheless, due to the convexity, we can lower bound the coverage guarantee directly without that information.

**Instance-wise worst case coverage.** Prop. 1 relies on lower bounding the worst-case (adversarial) $\tilde{\beta}_{n+1} = \Pr_{\epsilon_{n+1}}[s(\tilde{x}_{n+1} + \epsilon_{n+1}, y_{n+1}) \geq q]$, for the perturbed $\tilde{x}_{n+1} = x_{n+1} + \delta \in \mathcal{B}(x_{n+1})$ given $\beta_{n+1} := \Pr_{\epsilon_{n+1}}[s(x_{n+1} + \epsilon_{n+1}, y_{n+1}) \geq q]$. Here, $(x_{n+1}, y_{n+1})$ and $\epsilon_{n+1}$ are realizations of $(X_{n+1}, Y_{n+1})$ and $E_{n+1}$. This is conditional to $q$, and hence to $\mathcal{D}_n$ and $\mathcal{E}_n$. Formally, we define a binary classifier, $f(z) = \mathbb{I}[s(z, y_{n+1}) \geq q]$ and $g(z) = \mathbb{E}_{\epsilon_{n+1}}[f(z + \epsilon_{n+1})]$ for which we have $\beta_{n+1} := g(x_{n+1})$. Note that $\mathcal{X}$ is a convex subset of $\mathbb{R}^d$ and the score $s(\cdot, y)$ is continuous everywhere, therefore our classifier is measurable [19]. We can lower bound $\tilde{\beta}_{n+1} = \min_{\tilde{x}_{n+1} \in \mathcal{B}(x_{n+1})} g(\tilde{x}_{n+1})$ and therefore $g(\tilde{x}_{n+1})$ for the given $x_{n+1}$ using the existing (binary) classification certificates, e.g. Cohen et al. [7].

**Certified lower bound $\tilde{\beta}$.** A smoothing binary certificate computes the bound $\mathrm{c}^{\downarrow}[g(\cdot), x_{n+1}, \mathcal{B}]$ regardless of the original definition of $f$ – mechanics of the score function or model – and only as a function of the value $\beta = g(x_{n+1})$. We use the $\mathrm{c}^{\downarrow}[\beta, \mathcal{B}]$ notation, following Zargarbashi and Bojchevski [26]. For the known $x_{n+1}$, a (pointwise) certified lower bound on $g(\tilde{x}_{n+1}) : \tilde{x}_{n+1} \in \mathcal{B}(x_{n+1})$ is obtained by searching for the worst measurable binary function $h : \mathcal{X} \rightarrow \{0, 1\}$ in $\mathcal{H}$ (set of all measurable functions) such that $h$ has the same smooth output as $f$ at $x_{n+1}$. Formally:

$$\mathrm{c}^{\downarrow}[\beta, \mathcal{B}_r] = \min_{h \in \mathcal{H}} \Pr_{\epsilon}[h(\tilde{t} + \epsilon) = 1] \quad \text{s.t.} \quad \Pr_{\epsilon}[h(t + \epsilon) = 1] = \mathbb{E}_{\epsilon}[g(x_{n+1})] = \beta \qquad (4)$$

The pair $(t, \tilde{t})$ are called canonical points. Cohen et al. [7], Yang et al. [25] discuss this in detail. Intuitively, the optimization in Eq. 4 is translation (and in some cases rotation) invariant, and with the symmetries in the ball and the smoothing scheme, for any $x$, and $\tilde{x}$ we can use a fixed set of canonical points. For $\ell_p$ balls, and symmetric additive smoothing (including isotropic Gaussian noise) these points are one at the center, and the other at the edge (or vertex) of the ball; i.e. $t = [0, 0, \ldots, 0]$, and $\tilde{t} = [r, 0, \ldots, 0]$. For a detailed discussion also see section D.1 from Zargarbashi and Bojchevski [26]. Since the function $f$ itself is a feasible solution to Eq. 4, it is a valid lower bound for $g(x_{n+1})$.

The mean-constrained binary certificate in Eq. 4 is a common problem in the randomized smoothing literature. It is efficiently solvable and in many cases has a closed form solution. For the isotopic Gaussian smoothing with $\ell_2$ (and $\ell_1$) ball the lower bound is $\tilde{\beta} = \Phi_\sigma(\Phi_\sigma^{-1}(\beta) - r)$ where $\Phi_\sigma$ is the CDF of the Gaussian distribution $\mathcal{N}(0, \sigma)$ [17]. Using the recipe from Yang et al. [25], in § 3.1, we discuss how to compute $\mathrm{c}^{\downarrow}[p, \mathcal{B}]$ for other smoothing schemes.

**Lemma 1.** $\mathrm{c}^{\downarrow}[\beta, \mathcal{B}]$ *as the solution to Eq. 4 is convex and monotonically increasing w.r.t.* $\beta$.

We defer the proofs to § D.2. There we rigorously prove Lemma 1 directly from the definition of Eq. 4 via duality. Here we provide a sketch of an alternative proof that is insightful. Lee et al. [18] show that to solve Eq. 4, the space $\mathcal{X}$ can be divided to (finite or infinite) regions of constant likelihood ratio $\mathcal{R}_t = \{z : \frac{\Pr[z = \tilde{t} + \epsilon]}{\Pr[z = t + \epsilon]} = c_t\}$. If we can sort these regions in descending order w.r.t $c_t$, the problem reduces to the following linear program which is a fractional knapsack problem:

$$\mathrm{c}^{\downarrow}[\beta, \mathcal{B}] = \min_{h \in [0,1]^T} h^\top \cdot q \quad \text{s.t.} \quad h^\top \cdot p = \beta$$

where $T$ is number or regions, $h_t$ is the average value of $h(\boldsymbol{z})$ inside the region $\mathcal{R}_t$, $p_t = \Pr[\boldsymbol{t} + \boldsymbol{\epsilon} \in \mathcal{R}_t]$, $q_t = \Pr[\tilde{\boldsymbol{t}} + \boldsymbol{\epsilon} \in \mathcal{R}_t]$, and the vectors $\boldsymbol{h}, \boldsymbol{p}, \boldsymbol{q}$ gather all $h_t, p_t, q_t$'s (see Lee et al. [18] for the derivation). W.l.o.g assume that the $\boldsymbol{p}$, and $\boldsymbol{q}$ are sorted decreasingly w.r.t. $c_t$. The optimal solution is $\boldsymbol{h}^\star = [1, 1, \ldots, m, 0, \ldots, 0]$, for some $m \in (0, 1)$ which is to fill regions in order up to when $\boldsymbol{h} \cdot \boldsymbol{p}$ reaches $\beta$. Each index of $\boldsymbol{h}^\star$ is one region being filled and by setting $h_t^* = 1$ the $\boldsymbol{h} \cdot \boldsymbol{q}$, and $\boldsymbol{h} \cdot \boldsymbol{p}$ increase by $q_t$, and $p_t$. Therefore $c^\downarrow[\beta, \mathcal{B}] = \boldsymbol{h}^\star \cdot \boldsymbol{q}$ is a continuous piecewise linear function with slope of $q_t/p_t$ which is increasing across regions. A piecewise linear function with an increasing slope in each piece is convex. This convexity directly helps us to bound $\mathbb{E}[c^\downarrow[\beta, \mathcal{B}]] \geq c^\downarrow[\mathbb{E}[\beta], \mathcal{B}]$.

Note that in Prop. 1 the guarantee is over the coverage probability and independent of the setup; therefore, without any change, one can use it to make conformal regression robust. Regardless of the downstream task the certificate is always for binary classification. Furthermore, the result is not restricted to a specific scheme and can be used for any smoothing and perturbation ball (see § 3.1).

**Coverage distribution.** The expected coverage probability is itself a random variable with expectation higher than $1-\alpha$. Under mild assumptions, $\Pr[S_{n+1} \geq q \mid \mathcal{D}_n] \sim \mathrm{Beta}((1-\alpha)\cdot(n+1), \alpha(n+1))$ [2]. That is, for any given fixed calibration set, the coverage fluctuates around $1-\alpha$, with variance inversely proportional to the size of the calibration set. Since in practice we only have one calibration set, understanding this distribution, and its variance, is important. While we do not know the distribution of the robust coverage, we can compute a conservative estimate by convolving the CDF of $\mathrm{Beta}$ and the function $c^\downarrow[\cdot, \mathcal{B}]$ (see Fig. 3-left). Similarly, convexity helps to bound the mean of this new distribution as shown in Prop. 1. Note, our method does not take into account the distribution of the scores or inputs (unlike BinCP [26]) and as a result it is very conservative. In Fig. 3-middle we show comparison of our guaranteed lower-bound coverage and the empirical coverage under adversarial attack, highlighting that our guarantee accounts for significantly more damage.

**Maintaining** $1 - \alpha$ **coverage.** Prop. 1 says that under perturbation, the coverage guarantee of CP calibrated with $1 - \alpha$ over augmented inference decreases at most by $c^\downarrow[1 - \alpha, \mathcal{B}]$. A simple solution to attain $1 - \alpha$ robust coverage is to set the nominal coverage to a value $1 - \alpha'$ such that $c^\downarrow[1 - \alpha', \mathcal{B}] \geq 1 - \alpha$. In general, we can find $1 - \alpha'$ using binary search, however, from [26] we know that in smoothing schemes like Gaussian, we have $c^\uparrow[c^\downarrow[p, \mathcal{B}], \mathcal{B}^{-1}] = p$ (see § D.3, Lemma 2). Therefore, to attain $1 - \alpha$ robust coverage, we only need to set the threshold as the $c^\uparrow[1 - \alpha, \mathcal{B}^{-1}]$ quantile of the calibration scores (see Fig. 3-right).

### 3.1 Robust Conformal Sets with Randomized Smoothing of All Shapes and Sizes

Both RCP1, and BinCP work with any smoothing and ball $\mathcal{B}$. However, some binary certificates are given as a robust radius $r^*$ – the radius up to which the prediction remains the same, i.e. $c^\downarrow[p, \mathcal{B}_{r^*}] = 0.5$. Yang et al. [25] provide a recipe to compute the $r^*$ for general $\ell_p$ certificate under additive randomized smoothing. We tweak their "differential" method to derive probability bounds. We phrase Prop. 2 in a notation close to [25] and far from our own, however the takeaway is simple: in short we define $\Omega(p)$ such that $1/\Omega(p)$ encodes the minimum perturbation to make an infinitesimal increase in the worst case classifier with expected value $\beta$. We use line integral to find the $c^\uparrow$ for the worst

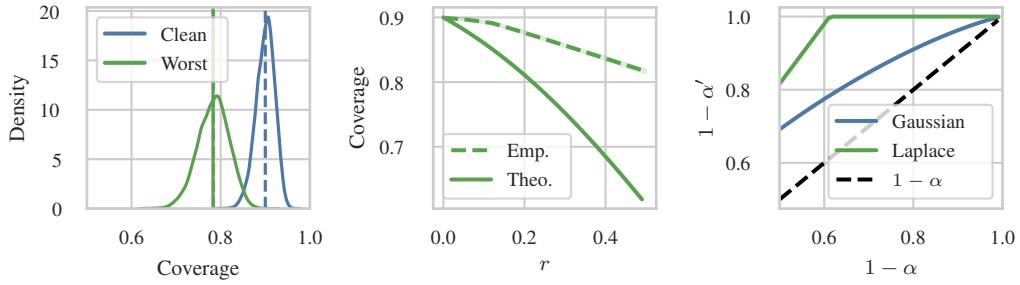

Figure 3: [Left] Samples from the $\mathrm{Beta}$ distribution of clean coverage, and worst-case coverage. The dashed line is the empirical average, and the overlapping solid line is $c^\downarrow[1 - \alpha, \mathcal{B}]$. [Middle] The empirical, and theoretical worst-case coverage. [Right] The robust $1 - \alpha'$ for two smoothing schemes.

case classifier at radius $r$; formally $\sup\{\overline{\beta} : \int_{\beta}^{\overline{\beta}} \frac{1}{\Omega(p)}\mathrm{d}p \leq r\}$. We can find this supremum either analytically (see § D.4) or via binary search using the existing closed forms provided by Yang et al. [25].

**Proposition 2.** *For a binary classifier $f(\boldsymbol{x})$, and an additive smoothing function $\xi$, Let $\mathcal{U} = \{\boldsymbol{x} : f(\boldsymbol{x}) = 1\}$ be the decision boundary and $\mathcal{U} - \boldsymbol{z}$ as the same set translated by $-\boldsymbol{z}$, such that $\boldsymbol{z}$ goes to the origin. Let $\xi(\mathcal{U}) = \mathrm{Pr}_{\boldsymbol{\epsilon} \sim \xi}[\boldsymbol{\epsilon} \in \mathcal{U}]$ be the expectation of the decision boundary under smoothing, and $\beta = \xi(\mathcal{U} - \boldsymbol{x}) = \mathbb{E}[f(\boldsymbol{x} + \boldsymbol{\epsilon})]$. Define:*

$$\Omega(p) := \sup_{\boldsymbol{\delta}:\|\boldsymbol{\delta}\|=1} \quad \sup_{\mathcal{U}\in\mathbb{R}^d:\xi(\mathcal{U})=p} \quad \lim_{r\to 0^+} \frac{\xi(\mathcal{U} - r\boldsymbol{\delta}) - p}{r}$$

*Assuming $\Omega(p)$ is strictly positive for $p \in [\mathrm{c}^{\downarrow}[\beta, \mathcal{B}_r], \mathrm{c}^{\uparrow}[\beta, \mathcal{B}_r]]$, and defining $F(\gamma) = \int_{\gamma}^{1/2} \frac{1}{\Omega(p)}\mathrm{d}p$:*

$$\mathrm{c}^{\uparrow}[\beta, \mathcal{B}_r] = \sup\{\beta' : F(\beta') \geq F(\beta) - r\} \tag{5}$$

*Similarly, we have $\mathrm{c}^{\downarrow}[\beta, \mathcal{B}_r] = \inf\{\beta' : F(\beta') \leq F(\beta) + r\}$.*

## 3.2 Extension to Conformal Risk Control

We use robustness certificates to define smoothing-based robust risk control for the first time. Let $\mathcal{C}_\lambda(\cdot)$ be a conformal set, where $\lambda \leq \lambda_{\max}$ controls the set size. For a risk function $\mathcal{L}(\boldsymbol{x}_i, y_i; \lambda) \in [a, b]$ that is right-continuous and non-increasing w.r.t. $\lambda$, if $\mathcal{L}(\boldsymbol{x}_i, y_i; \lambda_{\max}) \leq \alpha$, Angelopoulos et al. [1] show:

$$\mathbb{E}_{\mathcal{D}_{n+1}}[\mathcal{L}(\boldsymbol{x}_{n+1}, y_{n+1}; \lambda^*)] \leq \alpha \quad \text{for} \quad \lambda^* = \inf\{\lambda : \frac{\sum_{i=1}^n \mathcal{L}(\boldsymbol{x}_i, y_i; \lambda) + b}{n+1} \leq \alpha\}$$

Here $\alpha \in [a, b]$ is any user adjusted risk level. Similar to conformal prediction, we can also define a randomly augmented risk function $\mathcal{L}(\boldsymbol{x}_i + \boldsymbol{\epsilon}_i, y_i; \lambda)$. The noise does not break the exchangeability and therefore $\mathbb{E}[\mathcal{L}(\boldsymbol{x}_{n+1} + \boldsymbol{\epsilon}_{n+1}, y_{n+1}; \lambda^*)] \leq \alpha$ for the $\lambda^*$ computed on the randomly augmented calibration set. Due to the continuous nature of the risk function, we now use confidence certificates:

$$\mathrm{c}_c^{\uparrow}[\beta, \mathcal{B}] = \max_{h\in\mathcal{H}} \mathbb{E}[h(\tilde{\boldsymbol{x}}_{n+1})] \quad \text{s.t.} \quad \mathbb{E}[h(\boldsymbol{x}_{n+1})] \leq \beta \tag{6}$$

Here $h : \mathcal{X} \to [0, 1]$. Similarly, Eq. 6 can be efficiently solved, and for the Gaussian distribution it has a closed form solution of $b \cdot \Phi_\sigma(\Phi_\sigma^{-1}(\frac{\beta-a}{b-a}) + r) - a(1 - \Phi_\sigma(\Phi_\sigma^{-1}(\frac{\beta-a}{b-a}) + r))$. With $[a, b] = [0, 1]$ (e.g. for the false negative rate risk) the closed form is identical to the classification certificate [17].

## 4 Experiments

**Metrics and Baseline.** We evaluate average set size (lower is better), and empirical coverage (exceeding $1 - \alpha$ on average). Note that in RCPs the empirical coverage conservatively exceeds $1 - \alpha$ by increasing $r$. Under perturbation this decreases at worst to $1-\alpha$. As BinCP [26] outperforms other robust CP approaches [24, 27], we set it as our main comparison baseline. All recent smoothing-based RCPs return non-informative sets ($\mathcal{C}(\boldsymbol{x}) = \mathcal{Y}$) for low number of samples (e.g. $\leq 32$). Note that our main contribution is to return efficient sets with *one inference per input*; therefore we do not expect RCP1 to outperform BinCP for a large sample-rate. Our reported results are over 100 iterations with calibration set randomly sampled from the data. Further details are in § E, and the code is in our GitHub.

Since we certify the coverage guarantee (instead of scores), we can use the same *binary* certificate for both classification and regression tasks. We discuss the classification here, and defer the regression task to § E. The algorithm remains the same, only for the regression we use the absolute distance from the ground truth as the score. To the best of our knowledge, this is the first conformal regression certificate based on randomized smoothing.

**Classification.** We compare methods for the CIFAR10, and ImageNet datasets. We have two inference pipelines The original pipeline from BinCP, and CAS (computationally cheap setup): we use the ResNet models trained with noise augmentation from Cohen et al. [7]. Because of the model size, large sample-rates, although inefficient, are not unrealistic. We also evaluate on an alternative more expensive pipeline outlined by Carlini et al. [6]: the input is first denoised by a diffusion model and then classified by a vision transformer. For CIFAR-10 we combine a 50M-parameter diffusion model from Dhariwal and Nichol [9], with a `ViT-B/16` from Dosovitskiy et al. [10], pretrained

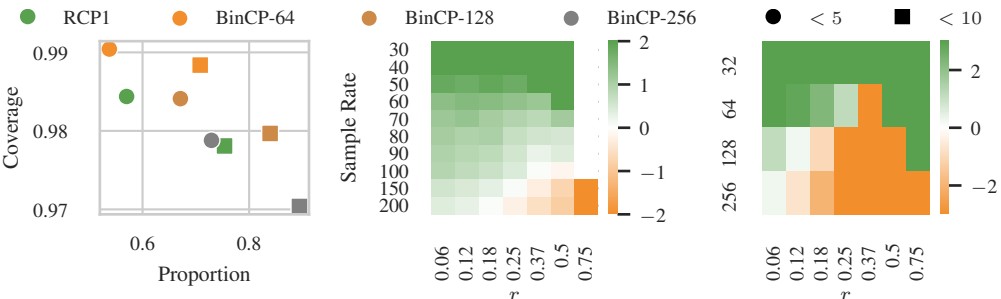

Figure 4: [Left] Proportion, and coverage of the prediction sets with $\leq 5$, and $\leq 10$ elements for the ViT model. [Middle] $|\mathcal{C}_{r,\text{BinCP}}| - |\mathcal{C}_{r,\text{RCP1}}|$ for the CIFAR-10 dataset with a ResNet model. [Right] ImageNet dataset and ViT models ($r = 0.25$). In all plots $\sigma = 0.5$ and RCP1 uses a single sample.

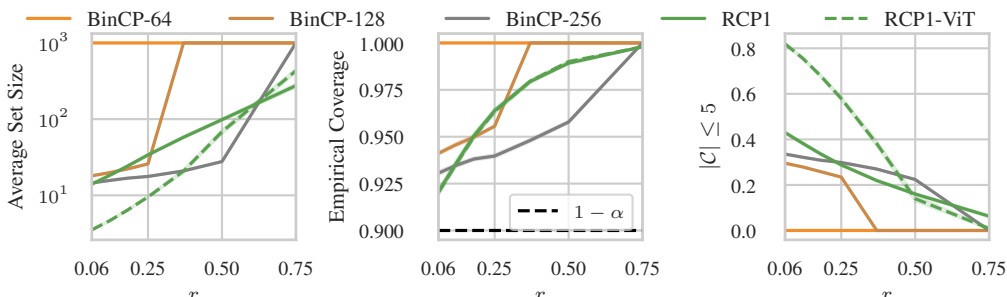

Figure 5: On ImageNet, cheap setup with $\sigma = 0.5$: [Left] Compares the average set size of BinCP with various sample rates to RCP1. [Middle] the empirical coverage. [Right] proportion of prediction sets with size less than 5 elements. Dashed line is the result of the ViT model with the same $\sigma$.

on ImageNet at $224 \times 224$ resolution and finetuned on CIFAR10 with 97.9% accuracy for the HuggingFace implementation. For ImageNet we use a 552M-parameter class-unconditional diffusion model followed by BEiT-L model (305M parameters) from Bao et al. [3] achieving 88.6% top-1 validation accuracy. We use the implementation provided by the timm library [23].

**Smaller set sizes.** Increasing the sample-rate (number of model forwards) in BinCP decreases the set size. Fig. 1-middle compares the set size and computation time for BinCP and RCP1 on the ImageNet dataset. Here, RCP1 shows similar set size to BinCP with 64 to 128 inferences per point. We also compare set size per radii for CIFAR-10 in Fig. 7, and for ImageNet in Fig. 5. A single inference over the larger pipeline (diffusion and ViT) for RCP1 takes significantly less time compared to the cheaper pipeline with enough samples for BinCP. Therefore we can easily achieve a considerably better set size with an unnoticeably more computation only by using a better model. In Fig. 4-middle, and right we compare BinCP and RCP1 in set size per sample rate (for BinCP) and radius for the CIFAR-10 and ImageNet datasets. Our complete comparison on this experiment is in § E. Note that it is significantly inefficient to run $\geq 100$ forwards passes per image on the ViT models. Additionally, we use the results in § 3.1 to show that the method works similarly for any smoothing scheme and threat model. For that we show the performance of BinCP (under two sample rates) and RCP1 for the $\ell_1$ ball under uniform smoothing distribution in Fig. 6-right.

For a dataset like ImageNet (with 1000 classes), the average set size alone is not a measure of usability. Consider a CP returning 50% singleton sets and $|\mathcal{Y}|$ for the rest, compared to a CP returning sets of size 100 for all inputs. Surely, the latter option is not usable even though it has smaller average set size. Hence, we also report the proportion of the prediction sets with less than 5 elements in Fig. 5-right (also see § E). This metric is only trustworthy if the we don't sacrifice the coverage in smaller sets. In Fig. 4-left we show that these sets have coverage larger than $1 - \alpha$.

**Small radii.** Jeary et al. [16], and Massena et al. [19] propose RCP using verifiers and Lipschitz constant of the network. Although their result is for one order of magnitude smaller radii (e.g. 0.02

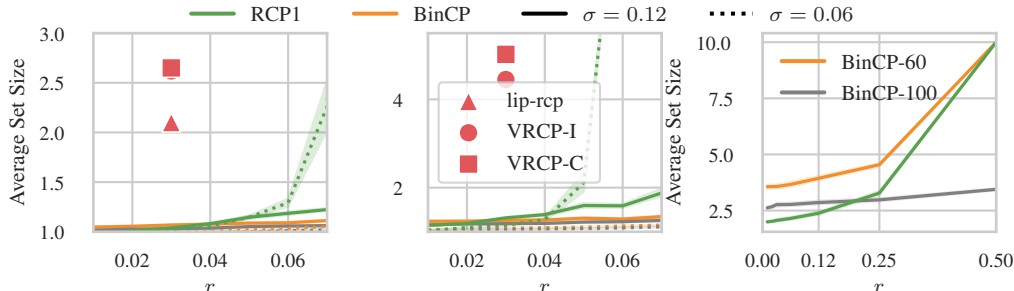

Figure 6: Performance on smaller radii in comparison with non-smoothing RCPs [16, 19] for CIFAR-10 and a ViT model. [Left] $1 - \alpha = 0.9$, and [Middle] $1 - \alpha = 0.95$. Smoothing is better. [Right] Performance of the methods for a Uniform-$\ell_1$ certificate for $\epsilon \sim \text{Uniform}[-1/\sqrt{3}, 1/\sqrt{3}]$.

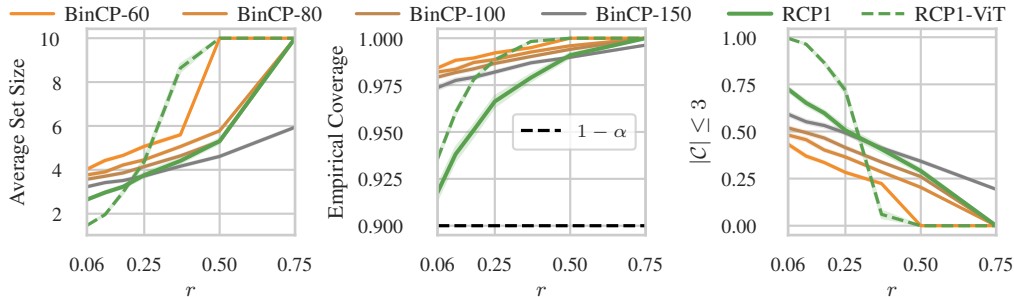

Figure 7: On CIFAR-10, ResNet model with $\sigma = 0.5$: [Left] The average set size compared to BinCP (with various sample rates), RCP1 [Middle] the empirical coverage, and [Right] proportion of sets with less than 3 labels. Dashed line shows the ViT model (with $\sigma = 0.25$).

instead of 0.25), their methods are efficient by using one forward per input. With RCP1 being the same in that metric, we compare with them in Fig. 6-left and middle reporting performance on smaller radii. Aside better performance, RCP1 has a black-box access and works for any model. Intuitively, as shown in Fig. 1-left, smooth (or augmented) inference is significantly more robust to perturbations.

**Time-comparison.** With $t_{\text{cert}}$, and $t_f$ as the time for computing bounds, and for the model's inference time, other smoothing based RCPs at best require $\mathcal{O}(n_{\text{mc}} \times \mathcal{D}_n \times t_f + t_{\text{cert}})$ for calibration where $n_{\text{mc}}$ is the number of MC samples. For each test point they also require $\mathcal{O}(n_{\text{mc}} \times t_f)$ time. RCP1 takes the same time as the normal model's inference plus an additional $\mathcal{O}(t_{\text{cert}})$ for calibration. Similarly, RCP1 takes $n_{\text{mc}}$ less memory compared to other smoothing RCPs. We show the runtime of the ViT pipeline for the used datasets in Table 1. Note that this is only the time to compute logits as the other processes (including certificates) are negligiable compared to it. The runtime of BinCP with a sample rate comparable to RCP1 is significantly high for large models like ViT; for instance, RCP1 and a comparable BinCP (with 128 samples on ImageNet) need $\sim$2',46", and $\sim$5h 55' to process 5000 images.

**Robust conformal risk control.** We use the model from Fischer et al. [11] on the CityScapes dataset [8] which is a scene segmentation task. We mask the regions where a target class (e.g. car) might be present. The error function is the false negative ratio (FNR) – the portion of the pixels from the

Table 1: Estimated runtimes (in `HH:MM:SS`) for 1000 inputs using an H-100 GPU. Results are scaled from a full experimental run assuming a linear cost in both the number of inputs and samples.

| Pipeline | Dataset | 1 Sample | 64 Samples | 128 Samples | 256 Samples |
|----------|---------|----------|------------|-------------|-------------|
| ViT | CIFAR-10 | 0:00:01 | 0:01:09 | 0:02:19 | 0:04:39 |
| | ImageNet | 0:00:33 | 0:35:30 | 1:11:00 | 2:22:01 |

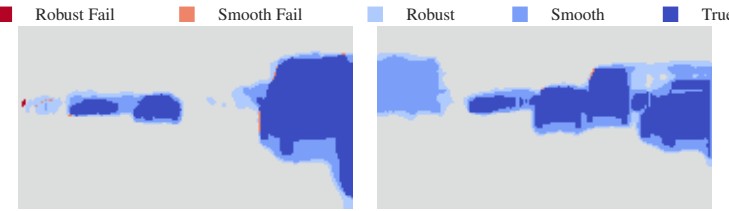

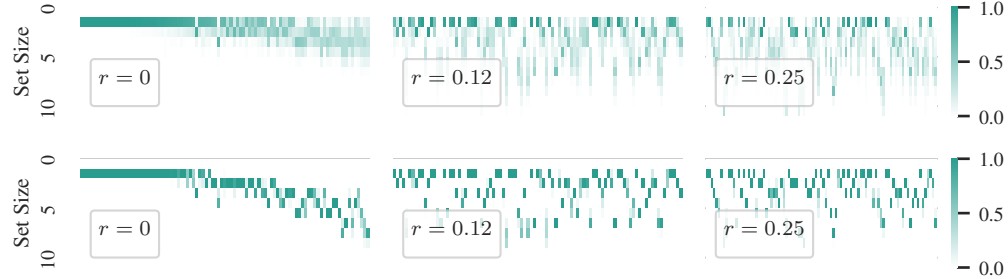

Figure 8: Performance of vanilla and robust risk control. From lighter to darker the colors are robust region, vanilla (non-robust) region, and the ground truth region. Here $\sigma = 0.25$, and $r = 0.06$.

Figure 9: Randomness in set sizes for a CIFAR-10 dataset and ResNet model. The $x$-axis sorts datapoints in a fixed order (same across all plots), and the $y$ axis shows the set size. The intensity of pixels shows the probability of a specific set size for that point over $\epsilon$. [Top] shows RCP1, and [Bottom] BinCP (150 samples). RCP1 has larger variance compared to BinCP.

target class that is not masked. We take the $\exp(f(\boldsymbol{x}))_y$ as the score of the class $y$, and we set the mask as $\mathbb{I}[\exp(f(\boldsymbol{x}))_y \geq 1 - \lambda]$. Note that here the classes could possibly overlap. We calibrate by finding a $\lambda$ that results in a FNR loss less than the user adjusted tolerable risk. So far, this is the first result for smoothing-based robust conformal risk control. Similar to RCP1, we first smooth the image data (one sample), then we compute the $\lambda$ that results in $c_c^{\downarrow}[\alpha, \mathcal{B}]$ risk. We report the results in Table 2, and show an example in Fig. 8.

**Limitation: Increased variance.** In Fig. 9, RCP1 shows considerably more randomness in the prediction sets compared to BinCP. This is essentially due to the random definition of the prediction set and the score function – the prediction set in RCP1 is a function of the random variable $\epsilon$. This randomness does not affect the final robust / vanilla coverage.

## 5 Conclusion

While offering small sets for larger radii, smoothing-based RCP methods need many forward passes per input. Instead, we show that noise-augmented inference combined with CP is inherently robust, and with that, we propose RCP1 which needs only one forward pass per input. Our approach returns sets with size similar to state of the art while nullifying the need of many MC samples. Prior smoothing RCPs provide their guarantee by lower bounding the scores, hence they need to estimate (some statistic about) the distribution of score for each individual input. Alternatively we only apply the lower bound on the coverage guarantee which is known in prior to be $1 - \alpha$.

Table 2: Risk and mask size for the Cityscapes dataset. Risk level is $0.15$, with $100$ calibration points. The variance is not over calibration sampling but over the images and $r = 0.06$.

| Class | Risk | Robust Risk | (True) Class Prop. | Mask Prop. | Robust Mask Prop. |
|---|---|---|---|---|---|
| Pedestrian | $0.1474 \pm 0.2797$ | $0.1111 \pm 0.2588$ | $0.0160 \pm 0.0279$ | $0.1891 \pm 0.1257$ | $0.2522 \pm 0.1312$ |
| Car | $0.1466 \pm 0.2582$ | $0.0833 \pm 0.2032$ | $0.0539 \pm 0.0545$ | $0.0832 \pm 0.0733$ | $0.1101 \pm 0.0807$ |

## Acknowledgements

We thank Guiliana Thomanek, and Jimin Cao for their feedback on our paper.

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

# A  Related Work

Gendler et al. [12] initially proposed robust CP resilient to adversarial examples (worst-case noise) without accounting for finite samples (asymptotically valid setup). Yan et al. [24] added finite sample correction and proposed a new score to return (set size) efficiency. Both mentioned works were using randomized smoothing and the mean of the smooth score to bound the worst case perturbations. Zargarbashi et al. [27] (CAS) proposed to use the CDF information of the smooth score – a more restrictive constraint and therefore returned smaller prediction sets. All of these methods required unrealistically expensive setup with $10^4$ MC samples to be able to return acceptably small sets. Zargarbashi and Bojchevski [26] (BinCP) defined a quantile-based score over the distribution of the smooth scores, that allowed same set size as CAS with orders of magnitude less samples (e.g. 200 would be sufficient). In contrast with all of the methods RCP1 works with a single augmented sample and without involving finite sample correction, which lies at the computation efficiency side of this trade-off.

Orthogonal to randomized smoothing, Jeary et al. [16], and Massena et al. [19] use verifiers and Lipschitz continuity of the networks to bound the score function. Their robust radii are one order of magnitude smaller than smoothing RCP, but instead they do not require many forward passes per input. In Fig. 6 we show that our approach (with the same computational efficiency) provides smaller prediction sets outperforming their results; plus, our approaches works for any black-box model. Notably all mentioned works provide robustness to the worst-case noise which is orthogonal to probabilistically robust CP Ghosh et al. [13].

# B  Additional Description of Figures

**Fig. 1-left.** We report how the empirical coverage of $\mathcal{C}_0$ (dashed lines) breaks for smooth prediction in BinCP and randomized augmented score in RCP1, compared to the vanilla conformal prediction (red dashed line). In all cases, we calibrate over clean calibration set and for radii $r$ we return the prediction set of $\tilde{\boldsymbol{x}}_{n+1}$ which is $\boldsymbol{x}_{n+1}$ perturbed with adversarial attacks. Specifically we use the `PGDSmooth` attack from [21] for smooth methods and conventional `PGD` attack for vanilla CP. Compared to `PGD`, the `PGDSmooth` attack performs stronger for smooth scores. The solid lines are shows the empirical coverage of $\mathcal{C}_r$ from BinCP and RCP1 on the same adversarial data. The main takeaway of the figure is to show the robustness of $\mathcal{C}_r$, and the inherent resilience of smooth and augmented inference to the adversarial (worst-case) noise. The result is for CIFAR-10 dataset, `ResNet` model and $\sigma = 0.5$.

**Fig. 1-middle.** We compared the robust set size of BinCP and RCP1; we plotted $|\mathcal{C}_{r,\text{BinCP}}| - |\mathcal{C}_{r,\text{RCP1}}|$ for which lower is better. The plot is for CIFAR-10 dataset, `ResNet` model and $\sigma = 0.5$.

**Fig. 1-right.** Each point is a computation of $\mathcal{C}_r$ shown both in time and set size. All times are divided by a single inference of the `ResNet` model. Here we evaluated two forward pipelines of cheaper `ResNet` model, and more time costly diffusion + vision transformer (as discussed in § 4). Both axis are log-scale and the plot is for the ImageNet dataset. Here $\sigma = 0.5$.

**Fig. 3-left.** We took samples from the Beta distribution of the coverage – each sample is then a number $\beta$. We computed the $\text{c}^{\downarrow}[\beta, \mathcal{B}]$ and draw a distribution of the new values. For the Beta distribution, we have $n = 200$, and $1 - \alpha = 0.9$. For the $\text{c}^{\downarrow}$ function we used $\sigma = 0.5$, and $r = 0.25$

**Fig. 3-middle.** We show the empirical coverage value under the PGDSmooth attack with $\sigma = 0.5$ over various radii. We use the same sigma to show the theoretical lower bound coverage.

**Fig. 3-right.** We reported the $\text{c}^{\downarrow}$ function for Gaussian and Laplace smoothing both with $\sigma = 0.5$. The plots are not empirical.

**Fig. 6-right.** Here we use the $\ell_1$ certificate from Yang et al. [25]. Here the smoothing scheme is $\epsilon = \text{Uniform}[-\sigma/\sqrt{3}, \sigma/\sqrt{3}]$

# C  More on Conformal Prediction

Our default score function in the manuscript is TPS (threshold prediction sets) where the score function is directly set to the softmax; $s(\boldsymbol{x}, y) = \text{Softmax}_y(f(\boldsymbol{x}))$ for the prediction model $f$. Another choice is to use the logits of the model as the conformity score $s(\boldsymbol{x}, y) = f(\boldsymbol{x})_y$. Similar to BinCP we are

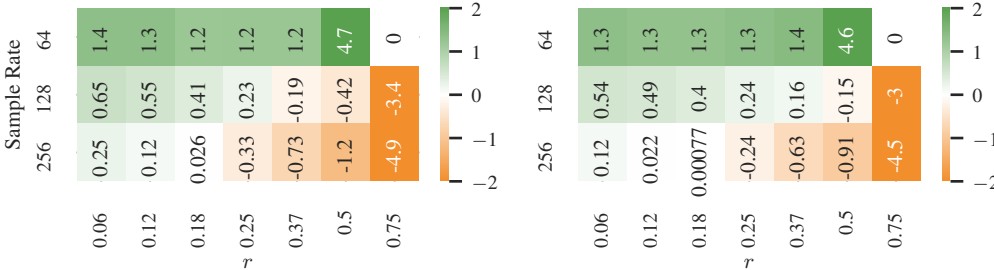

Figure 10: Comparison of BinCP and RCP1 for [Left] TPS, and [Right] APS score function on CIFAR-10 dataset with $\sigma = 0.9$.

Table 3: Set size of RCP1 for TPS and APS score across radii ($r$) and target coverage guarantees.

| $r$ | Coverage | TPS | | APS | |
|---|---|---|---|---|---|
| | | Avg Set Size | Emp. Cov. | Avg Set Size | Emp. Cov. |
| 0.06 | 0.85 | $2.17 \pm 0.02$ | $0.88 \pm 0.00$ | $2.52 \pm 0.03$ | $0.88 \pm 0.00$ |
| | 0.90 | $2.70 \pm 0.03$ | $0.92 \pm 0.00$ | $2.98 \pm 0.02$ | $0.92 \pm 0.00$ |
| | 0.95 | $3.74 \pm 0.01$ | $0.97 \pm 0.00$ | $3.91 \pm 0.04$ | $0.97 \pm 0.00$ |
| 0.12 | 0.85 | $2.44 \pm 0.03$ | $0.90 \pm 0.00$ | $2.76 \pm 0.02$ | $0.90 \pm 0.00$ |
| | 0.90 | $2.93 \pm 0.04$ | $0.94 \pm 0.00$ | $3.23 \pm 0.01$ | $0.94 \pm 0.00$ |
| | 0.95 | $3.96 \pm 0.06$ | $0.97 \pm 0.00$ | $4.07 \pm 0.04$ | $0.97 \pm 0.00$ |
| 0.18 | 0.85 | $2.70 \pm 0.04$ | $0.92 \pm 0.00$ | $2.99 \pm 0.01$ | $0.92 \pm 0.00$ |
| | 0.90 | $3.25 \pm 0.05$ | $0.95 \pm 0.00$ | $3.44 \pm 0.03$ | $0.95 \pm 0.00$ |
| | 0.95 | $4.48 \pm 0.09$ | $0.98 \pm 0.00$ | $4.48 \pm 0.02$ | $0.98 \pm 0.00$ |
| 0.25 | 0.85 | $3.03 \pm 0.01$ | $0.94 \pm 0.00$ | $3.33 \pm 0.04$ | $0.94 \pm 0.00$ |
| | 0.90 | $3.70 \pm 0.02$ | $0.97 \pm 0.00$ | $3.89 \pm 0.03$ | $0.97 \pm 0.00$ |
| | 0.95 | $4.81 \pm 0.01$ | $0.99 \pm 0.00$ | $4.82 \pm 0.04$ | $0.99 \pm 0.00$ |
| 0.37 | 0.85 | $3.62 \pm 0.06$ | $0.96 \pm 0.00$ | $3.91 \pm 0.01$ | $0.97 \pm 0.00$ |
| | 0.90 | $4.48 \pm 0.03$ | $0.98 \pm 0.00$ | $4.55 \pm 0.09$ | $0.98 \pm 0.00$ |
| | 0.95 | $6.24 \pm 0.14$ | $0.99 \pm 0.00$ | $6.49 \pm 0.04$ | $1.00 \pm 0.00$ |
| 0.50 | 0.85 | $4.51 \pm 0.03$ | $0.98 \pm 0.00$ | $4.55 \pm 0.11$ | $0.98 \pm 0.00$ |
| | 0.90 | $5.32 \pm 0.04$ | $0.99 \pm 0.00$ | $5.34 \pm 0.03$ | $0.99 \pm 0.00$ |
| | 0.95 | $10.00 \pm 0.00$ | $1.00 \pm 0.00$ | $10.00 \pm 0.00$ | $1.00 \pm 0.00$ |
| 0.75 | 0.85 | $6.28 \pm 0.19$ | $0.99 \pm 0.00$ | $6.31 \pm 0.11$ | $0.99 \pm 0.00$ |
| | 0.90 | $10.00 \pm 0.00$ | $1.00 \pm 0.00$ | $10.00 \pm 0.00$ | $1.00 \pm 0.00$ |
| | 0.95 | $10.00 \pm 0.00$ | $1.00 \pm 0.00$ | $10.00 \pm 0.00$ | $1.00 \pm 0.00$ |

using a single binary certificate which do not rely on bounded score function. Another conformal prediction method called as adaptive prediction sets (APS) uses the accumulated softmax up to label $y$ as the conformity score; formally $s(\boldsymbol{x}, y) = -[\pi(\boldsymbol{x}, y) \cdot u + \sum_{k=1}^{|\mathcal{Y}|} \pi(\boldsymbol{x}, y) \cdot \mathbb{I}[\pi(\boldsymbol{x}, y_k) > \pi(\boldsymbol{x}, y)]]$ where $\pi(\boldsymbol{x}, y) = \text{Softmax}_y(f(\boldsymbol{x}))$ and $u \sim \text{Uniform}[0, 1]$. While this score results in larger sets, it increases adaptivity – approximate conditional coverage.

We report the result using these score functions in Fig. 10 (comparison of BinCP and RCP1 for each score) and Table 3 (the set size of RCP1 for both score functions). As expected, similar trend as TPS is observed for APS as well.

Same as BinCP we also do not need the score function to be bounded. However in the end, using an unbounded score function (like using logits directly) did not show to improve over the existing APS and TPS.

# D Supplementary to Theory

## D.1 Robust Conformal Prediction Guarantee

The guarantee in Eq. 2 doesn't take the randomness in score function and prediction set into account. This is while many conformal scores have a random variable inside, for instance APS [20] multiplies the probability of each class by a uniform random value to break the ties (and enable the exact $1 - \alpha$ coverage). The original guarantee taken from [13, 27] is:

$$\Pr_{\mathcal{D}_n, \boldsymbol{x}_{n+1} \sim \mathcal{D}} [y_{n+1} \in \mathcal{C}_{\mathcal{B}}(\tilde{\boldsymbol{x}}_{n+1}), \forall \tilde{\boldsymbol{x}}_{n+1} \in \mathcal{B}(\boldsymbol{x}_{n+1})] \geq 1 - \alpha$$

$$\equiv \Pr_{\mathcal{D}_n, \boldsymbol{x}_{n+1} \sim \mathcal{D}} \left[ \inf_{\forall \tilde{\boldsymbol{x}}_{n+1} \in \mathcal{B}(\boldsymbol{x}_{n+1})} \mathbb{I}[y_{n+1} \in \mathcal{C}_{\mathcal{B}}(\tilde{\boldsymbol{x}}_{n+1})] = 1 \right] \geq 1 - \alpha$$

This formulation fails to capture the stochasticity in the score function (and hence in the prediction set). Since with a very small probability to miscover a point (which is non-zero in methods like APS) the indicator evaluates to false. The worst-case indicator $\inf_{\forall \tilde{\boldsymbol{x}}_{n+1} \in \mathcal{B}(\boldsymbol{x}_{n+1})} \mathbb{I}[y_{n+1} \in \mathcal{C}_{\mathcal{B}}(\tilde{\boldsymbol{x}}_{n+1}) = 1]$ becomes zero whenever there exists any probability that $\boldsymbol{x}_{n+1}$ is not covered. Consider the non-robust case where no perturbations are applied; i.e., evaluating the coverage guarantee of standard conformal prediction. Using the APS score function and shrinking the perturbation space to an infinitesimal ball $\mathcal{B}_r$ as $r \to 0^+$ (and therefore $\tilde{\boldsymbol{x}}_{n+1} \to \boldsymbol{x}_{n+1}$), the coverage probability should exceed $1 - \alpha$, since APS already satisfies this guarantee. However many of the datapoints that have a small probability to exclude the true class from the prediction set will not pass the worst-case indicator. Consider a datapoint with one hot conditional probability, still the top label will be in the prediction set with probability $-q$ where $q$ is the threshold.

While the shortcoming in Eq. 2 excludes any CP with randomness in the score, still previous smoothing-based RCP methods (at least in assymptotically valid setup) satisfy its conditions independent of the score function used. This is since all previous methods systematically remove all the randomness from the score and return a deterministic prediction set. Given any base score function, these methods defined their own score as a statistic (e.g. mean, or quantile) over the distribution of the base score on $\boldsymbol{x} + \boldsymbol{\epsilon}$. As the distribution already includes the inherent randomness in the base score itself, the statistics like mean [12, 27] and the quantile [26] are deterministic (excluding any randomness). This is an orthogonal to the probabilistic nature of estimating these statistics from Monte Carlo samples (the validity of confidence intervals). As a result the final set based on these scores exclude the inherent randomness of the base score function.

## D.2 Proofs

*Proof of Lemma 1.* The Lagrangian form of Eq. 4 is:

$$\mathcal{L}(\beta, \lambda) = \min_{h \in \{0,1\}^{|\mathcal{X}|}} \Pr_{\boldsymbol{\epsilon}}[h(\tilde{\boldsymbol{t}} + \boldsymbol{\epsilon}) = 1] - \lambda \left( \Pr_{\boldsymbol{\epsilon}}[h(\boldsymbol{t} + \boldsymbol{\epsilon}) = 1] - \beta \right)$$

$$= \min_{h \in \{0,1\}^{|\mathcal{X}|}} E_{\boldsymbol{z} \sim q}[h(\boldsymbol{z})] - \lambda \cdot E_{\boldsymbol{z} \sim p}[h(\boldsymbol{z}) - \beta] = \lambda \cdot \beta + \min_{h \in \{0,1\}^{|\mathcal{X}|}} E_{\boldsymbol{z} \sim q}[h(\boldsymbol{z})] - E_{\boldsymbol{z} \sim p}[h(\boldsymbol{z})]$$

$$= \lambda \cdot \beta + \min_{h \in \{0,1\}^{|\mathcal{X}|}} \int_{\mathcal{X}} (q(\boldsymbol{z}) - \lambda \cdot p(\boldsymbol{z})) \cdot h(\boldsymbol{z}) \mathrm{d}\boldsymbol{z}$$

where $p$ and $q$ are smoothing distributions centered at $\boldsymbol{t}$ and $\tilde{\boldsymbol{t}}$ respectively. The worst classifier (the minimizer of the problem) can be derived as follows

$$h(\boldsymbol{z}) = \begin{cases} 0 & \text{if} \quad q(\boldsymbol{z}) - \lambda \cdot p(\boldsymbol{z}) \geq 0 \\ 1 & \text{otherwise} \end{cases}$$

Intuitively, to minimize the term $\int_{\mathcal{X}} (q(\boldsymbol{z}) - \lambda \cdot p(\boldsymbol{z})) \cdot h(\boldsymbol{z}) \mathrm{d}\boldsymbol{z}$ we look at each point independently. For each point if the term $(q(\boldsymbol{z}) - \lambda \cdot p(\boldsymbol{z}))$ is positive we cancel it by $h = 0$ and if negative we keep it to decrease the total integral value. The resulting dual with the dual variable $\lambda \geq 0$ is then:

$$\mathcal{L}(\beta, \lambda) = \lambda \cdot \beta + \int \min\{0, p(\boldsymbol{z}) - \lambda \cdot q(\boldsymbol{z}) \mathrm{d}\boldsymbol{z}\} = \lambda \cdot \beta + l(\lambda)$$

Here $l(\lambda) := \int \min\{0, p(\boldsymbol{z}) - \lambda \cdot q(\boldsymbol{z}) \mathrm{d}\boldsymbol{z}\}$ is only a function of $\lambda$. Maximizing over $\lambda$ we get the optimal dual soltuion which equals the optimal primal since it was shown that strong duality holds [28]:

$$\mathrm{c}^{\downarrow}[\beta, \mathcal{B}] = \max_{\lambda} \lambda \cdot \beta + l(\lambda)$$

Which is pointwise maximum of affine functions and therefore convex in $\beta$.

The monotonicity w.r.t. $\beta$ directly follows from the definition. By increasing $\beta$ the feasible space reduces to a nested subset of the previous problem which means that the solution will be greater than or equal to the original solution. $\qquad\square$

### D.3 Choosing the conservative $1 - \alpha'$

In general to obtain $1 - \alpha$ robust coverage guarantee, we should choose the nominal $1 - \alpha'$ in Alg. 1 such that $\mathrm{c}^{\downarrow}[1 - \alpha, \mathcal{B}] \geq 1 - \alpha$. This nominal probability can be found via binary search due to the non-decreasing nature of $\mathrm{c}^{\downarrow}$. But in many cases including the Gaussian distribution, where the canonical points can be used interchangeably (choosing $(\boldsymbol{t}, \tilde{\boldsymbol{t}})$ as the pair of clean, and noisy canonical points doesn't differ from the opposite $(\tilde{\boldsymbol{t}}, \boldsymbol{t})$), the following lemma allows us to set the $1 - \alpha' = \mathrm{c}^{\uparrow}[1 - \alpha, \mathcal{B}^{-1}]$.

**Lemma 2.** *If for a smoothing scheme, and a perturbation ball $\mathcal{B}$, canonical points $\boldsymbol{t}$, and $\tilde{\boldsymbol{t}}$ can be used interchangeably; then we have $\mathrm{c}^{\uparrow}[\mathrm{c}^{\downarrow}[p, \mathcal{B}], \mathcal{B}^{-1}] = p$. Using the canonical points interchangeably means that for $\mathrm{c}^{\uparrow}$, and (similarly $\mathrm{c}^{\downarrow}$) both of the optimizations*

$$\max_{h \in \mathcal{H}} \Pr_{\boldsymbol{\epsilon}}[h(\tilde{\boldsymbol{t}} + \boldsymbol{\epsilon}) = 1] \quad s.t. \quad \Pr_{\boldsymbol{\epsilon}}[h(\boldsymbol{t} + \boldsymbol{\epsilon}) = 1] = p$$

*and*

$$\max_{h \in \mathcal{H}} \Pr_{\boldsymbol{\epsilon}}[h(\boldsymbol{t} + \boldsymbol{\epsilon}) = 1] \quad s.t. \quad \Pr_{\boldsymbol{\epsilon}}[h(\tilde{\boldsymbol{t}} + \boldsymbol{\epsilon}) = 1] = p$$

*yield the same solution.*

*Proof.* The term $\mathrm{c}^{\uparrow}[\mathrm{c}^{\downarrow}[p, \mathcal{B}], \mathcal{B}^{-1}]$ is expressed as the following optimization problem:

$$p_{\text{high}}^* = \max_{h \in \mathcal{H}} \Pr_{\boldsymbol{\epsilon}}[h(\tilde{\boldsymbol{t}} + \boldsymbol{\epsilon}) = 1] \quad \text{s.t.} \quad \Pr_{\boldsymbol{\epsilon}}[h(\boldsymbol{t} + \boldsymbol{\epsilon}) = 1] = p_{\text{low}}^* \tag{7}$$

$$p_{\text{low}}^* = \min_{h' \in \mathcal{H}} \Pr_{\boldsymbol{\epsilon}}[h'(\tilde{\boldsymbol{t}} + \boldsymbol{\epsilon}) = 1] \quad \text{s.t.} \quad \Pr_{\boldsymbol{\epsilon}}[h'(\boldsymbol{t} + \boldsymbol{\epsilon}) = 1] = \Pr_{\boldsymbol{\epsilon}}[f(\boldsymbol{t} + \boldsymbol{\epsilon})] = p$$

We swap $\tilde{\boldsymbol{t}}$, and $\boldsymbol{t}$ since we can use the canonical points interchangeably. We have

$$p_{\text{high}}^* = \max_{h \in \mathcal{H}} \Pr_{\boldsymbol{\epsilon}}[h(\boldsymbol{t} + \boldsymbol{\epsilon}) = 1] \quad \text{s.t.} \quad \Pr_{\boldsymbol{\epsilon}}[h(\tilde{\boldsymbol{t}} + \boldsymbol{\epsilon}) = 1] = p_{\text{low}}^*$$

$h_{\text{low}}^*$ as the solution to the inner problem in Eq. 7 (defining $p_{\text{low}}^*$) is a feasible solution to the outer optimization (the first line); therefore

$$p_{\text{high}}^* = \max_{h \in \mathcal{H}} \Pr_{\boldsymbol{\epsilon}}[h(\boldsymbol{t} + \boldsymbol{\epsilon}) = 1] \geq \Pr_{\boldsymbol{\epsilon}}[h_{\text{low}}^*(\boldsymbol{t} + \boldsymbol{\epsilon}) = 1] = \Pr_{\boldsymbol{\epsilon}}[f(\boldsymbol{t} + \boldsymbol{\epsilon})] = p$$

Both functions $\mathrm{c}^{\uparrow}$, and $\mathrm{c}^{\downarrow}$ (and therefore both minimization and maximization) are non-decreasing to the value in their constraint. Assuming $p_{\text{high}}^* > p$ ($p_{\text{high}}^* \neq p$), we have $p = \mathrm{c}^{\uparrow}[p_{\text{low}}', \mathcal{B}^{-1}]$ that $p_{\text{low}}' < p_{\text{low}}^*$ (due to non-decreasing nature of $\mathrm{c}^{\uparrow}$). We have

$$p = \max_{h \in \mathcal{H}} \Pr_{\boldsymbol{\epsilon}}[h(\boldsymbol{t} + \boldsymbol{\epsilon}) = 1] \quad \text{s.t.} \quad \Pr_{\boldsymbol{\epsilon}}[h(\tilde{\boldsymbol{t}} + \boldsymbol{\epsilon}) = 1] = p_{\text{low}}'$$

with a maximizer function $h_{\text{high}}'$; i.e. $p = \Pr_{\boldsymbol{\epsilon}}[h_{\text{high}}'(\boldsymbol{t} + \boldsymbol{\epsilon}) = 1]$. We rewrite inner problem in Eq. 7

$$p_{\text{low}}^* = \min_{h' \in \mathcal{H}} \Pr_{\boldsymbol{\epsilon}}[h'(\tilde{\boldsymbol{t}} + \boldsymbol{\epsilon}) = 1] \quad \text{s.t.} \quad \Pr_{\boldsymbol{\epsilon}}[h'(\boldsymbol{x} + \boldsymbol{\epsilon}) = 1] = \Pr_{\boldsymbol{\epsilon}}[f(\boldsymbol{t} + \boldsymbol{\epsilon})] = p$$

The maximizer function $h_{\text{high}}'$ satisfies the constraint, and therefore $p_{\text{low}}^* < p_{\text{low}}'$ which is a contradiction. Therefore $p_{\text{high}}^* = p$. $\qquad\square$

## D.4 Lower and Upper Bounds for All Shapes and Sizes

**Lemma 3.** *For a binary classifier $f(\boldsymbol{x})$, let $g(\boldsymbol{x}) = \Pr_{\boldsymbol{\epsilon}}[f(\boldsymbol{x} + \boldsymbol{\epsilon}) = 1]$ and*

$$g(\tilde{\boldsymbol{x}}) \geq \mathrm{c}_g^{\downarrow}[p, \mathcal{B}] := \min_{h \in \mathcal{H}} \Pr_{\boldsymbol{\epsilon}}[h(\tilde{\boldsymbol{x}} + \boldsymbol{\epsilon}) = 1] \quad s.t. \ \Pr_{\boldsymbol{\epsilon}}[h(\boldsymbol{x} + \boldsymbol{\epsilon}) = 1] = g(\boldsymbol{x}) = p$$

*Similarly let*

$$g(\tilde{\boldsymbol{x}}) \leq \mathrm{c}_g^{\uparrow}[p, \mathcal{B}] := \max_{h \in \mathcal{H}} \Pr_{\boldsymbol{\epsilon}}[h(\tilde{\boldsymbol{x}} + \boldsymbol{\epsilon}) = 1] \quad s.t. \ \Pr_{\boldsymbol{\epsilon}}[h(\boldsymbol{x} + \boldsymbol{\epsilon}) = 1] = g(\boldsymbol{x}) = p$$

*Both be obtainable at the same canonical points. We have $\mathrm{c}_g^{\uparrow}[p, \mathcal{B}] = 1 - \mathrm{c}_g^{\downarrow}[1 - p, \mathcal{B}]$.*

*Proof.* For simpler notation let $\overline{g}(\boldsymbol{x}) = \mathrm{c}_g^{\uparrow}[g(\boldsymbol{x}), \mathcal{B}]$, $\underline{g}(\boldsymbol{x}) = \mathrm{c}_g^{\downarrow}[g(\boldsymbol{x}), \mathcal{B}]$, then we have

$$1 - \overline{g}(\boldsymbol{x}) = 1 - \max_{h \in \mathcal{H}} \Pr_{\boldsymbol{\epsilon}}[h(\tilde{\boldsymbol{x}} + \boldsymbol{\epsilon}) = 1]$$

$$\text{Let } h'(\boldsymbol{x}) = 1 - h(\boldsymbol{x}) \text{ then}$$

$$= 1 - \max_{h' \in \mathcal{H}} \Pr_{\boldsymbol{\epsilon}}[1 - h'(\tilde{\boldsymbol{x}} + \boldsymbol{\epsilon}) = 1] = \min_{h' \in \mathcal{H}} \Pr_{\boldsymbol{\epsilon}}[h'(\tilde{\boldsymbol{x}} + \boldsymbol{\epsilon}) = 1]$$

The constraint also translates similarly

$$\Pr_{\boldsymbol{\epsilon}}[h(\boldsymbol{x} + \boldsymbol{\epsilon}) = 1] = \Pr_{\boldsymbol{\epsilon}}[1 - h'(\boldsymbol{x} + \boldsymbol{\epsilon}) = 1] = 1 - \Pr_{\boldsymbol{\epsilon}}[h'(\boldsymbol{x} + \boldsymbol{\epsilon}) = 1] = 1 - p$$

And the new problem is by definition same as $1 - \mathrm{c}^{\downarrow}[1 - p, \mathcal{B}]$. $\qquad\square$

**Certified Upper and Lower bounds for all Shapes and Sizes.** In Prop. 2 we rephrased the Theorem 4.1 from Yang et al. [25] to return the upper bound probability instead of the robust radius. Here we prove Prop. 2, using the original proof from Yang et al. [25].

Let $g(\boldsymbol{x}) := \mathbb{E}_{\boldsymbol{\epsilon}}[f(\boldsymbol{x} + \boldsymbol{\epsilon})]$ for any binary decision function $f$ and any $\boldsymbol{\epsilon} \sim \xi$ where $\xi(\boldsymbol{x}) \propto \exp(-\psi(\boldsymbol{x}))$. If $g(\cdot)$ is continuous in $\mathcal{X}$, for any point $\tilde{\boldsymbol{x}} = \boldsymbol{x} + \boldsymbol{\delta}$ one can compute the $g(\boldsymbol{x} + \tilde{\boldsymbol{\delta}})$ through line integral as

$$g(\tilde{\boldsymbol{x}}) = g(\boldsymbol{x} + \boldsymbol{\delta}) = g(\boldsymbol{x}) + \int_0^r \frac{\mathrm{d}}{\mathrm{dt}}[g(\boldsymbol{x} + t \cdot \boldsymbol{\delta}')]\mathrm{dt}$$

where $\boldsymbol{\delta}' = \frac{\boldsymbol{\delta}}{\|\boldsymbol{\delta}\|}$ is the unit vector in the same direction as $\boldsymbol{\delta}$ and $r := \|\boldsymbol{\delta}\|$. In other words, we add all the infinitisimal changes on the path from $\boldsymbol{x}$ to $\tilde{\boldsymbol{x}}$ to compute the value of $g(\tilde{\boldsymbol{x}})$ given $g(\boldsymbol{x})$.

With the decision boundary $\mathcal{U} = \{\boldsymbol{x} : f(\boldsymbol{x}) = 1\}$, and $\mathcal{U} - \boldsymbol{z}$ as the decision boundary translated by $-\boldsymbol{z}$, consider the following function

$$\Omega(p) := \sup_{\boldsymbol{\delta}:\|\boldsymbol{\delta}\|=1} \ \sup_{\mathcal{U} \in \mathbb{R}^d:\xi(\mathcal{U})=p} \ \lim_{r \to 0^+} \frac{\xi(\mathcal{U} - r\boldsymbol{\delta}) - p}{r}$$

Proposition F.8 from Yang et al. [25] show that anywhere in $\mathcal{X}$, we have $\frac{\mathrm{d}}{\mathrm{dt}}[g(\boldsymbol{z})] \leq \Omega(g(\boldsymbol{z}))$. This means that one can upper bound the growth of $g(\boldsymbol{x})$ while shifted by $\boldsymbol{\delta}$ by integrating over $\Omega(g(\boldsymbol{z}))$ instead. For easier notation let $h(t) = g(\boldsymbol{x} + t \cdot \boldsymbol{\delta}')$ which implies $h'(t) = \frac{\mathrm{d}}{\mathrm{dt}}h(t) \leq \Omega(h(t))$. Notably, the function $\Omega(p)$ is always non-negative. See the equivalent Definition H.12 from Yang et al. [25] where the function is denoted as $\Phi(p)$ and it is written as an expectation of a maximum over a value that is always non-negative. For positive (non zero) $\Omega(h(t))$, including but not limited to $h(t) \leq 1/2$ (see the definition of the function in Appendix F from Yang et al. [25]) It follows:

$$\frac{h'(t)}{\Omega(h(t))} \leq 1 \Rightarrow \int_0^r \frac{h'(t)}{\Omega(h(t))}\mathrm{dt} \leq \int_0^r 1\mathrm{dt} = r$$

We set $u = h(t) \Rightarrow \mathrm{d}u = h'(t)\mathrm{dt}$ which implies

$$\Pi(u) = \int_0^r \frac{h'(t)}{\Omega(h(t))}\mathrm{dt} = \int_{u=h(0)}^{u=h(r)} \frac{1}{\Omega(u)}\mathrm{d}u \leq r$$

Here $h(0) = g(\boldsymbol{x} + 0 \cdot \boldsymbol{\delta}') = g(\boldsymbol{x}) = \beta$, and $h(0) = g(\boldsymbol{x} + r \cdot \boldsymbol{\delta}') = g(\tilde{\boldsymbol{x}}) = \bar{\beta}$. With the reference function $F(\gamma) = \int_\gamma^{1/2} \frac{1}{\Omega(p)} \mathrm{d}p$ we have

$$r \geq \int_\beta^{\bar{\beta}} \frac{1}{\Omega(p)} \mathrm{d}p = \int_\beta^{1/2} \frac{1}{\Omega(p)} \mathrm{d}p - \int_{\bar{\beta}}^{1/2} \frac{1}{\Omega(p)} \mathrm{d}p = F(\beta) - F(\bar{\beta}) \Rightarrow F(\beta) - F(\bar{\beta}) \leq r$$

which is $F(\bar{\beta}) \geq F(\beta) - r$.

The authors already computed $\Omega(p)$ (in their paper it is called as $\Phi$) for following several distributions, including:

- Isotropic Gaussian smoothing against $\ell_2$ ball ($\sigma = 1$): $\Omega(u) = \Phi'(\Phi^{-1}(1 - u))$ which implies:

$$\Pi_{\text{Gaussian}}(u) = \int_\beta^{\bar{\beta}} \frac{1}{\Omega(u)} \mathrm{d}u = \int_\beta^{\bar{\beta}} \frac{1}{\Phi'(\Phi^{-1}(1 - u))} \mathrm{d}u$$

  With $c = \Phi^{-1}(1 - u)$ we have $\mathrm{d}p = -\Phi'(c)\mathrm{d}c$, and $\Phi'(c) = \Phi'(\Phi^{-1}(1 - u))$. Therefore

$$\int_\beta^{\bar{\beta}} \frac{1}{\Phi'(\Phi^{-1}(1 - u))} \mathrm{d}u = \int_{\Phi^{-1}(1-\beta)}^{\Phi^{-1}(1-\bar{\beta})} -\mathrm{d}u = \Phi^{-1}(1 - \beta) - \Phi^{-1}(1 - \bar{\beta}) \leq r$$
$$\Rightarrow \Phi^{-1}(1 - \bar{\beta}) \geq \Phi^{-1}(1 - \beta) - r \Rightarrow 1 - \Phi^{-1}(\bar{\beta}) \geq 1 - \Phi^{-1}(\beta) - r$$
$$\Rightarrow \Phi^{-1}(\bar{\beta}) \leq \Phi^{-1}(\beta) + r \rightarrow \bar{\beta} \leq \Phi(\Phi^{-1}(\beta) + r)$$

  Which completely aligns with the aforementioned closed form $\ell_2$ certificate.

- Laplace smoothing against $\ell_1$ ball ($\sigma = \sqrt{2}\lambda$): $\Omega(u) = \frac{u}{\lambda}$ which implies:

$$\Pi_{\text{Laplace}}(u) = \int_\beta^{\bar{\beta}} \frac{1}{\Omega(u)} \mathrm{d}u = \int_\beta^{\bar{\beta}} \lambda \frac{1}{u} \mathrm{d}u = \lambda \log \frac{\bar{\beta}}{\beta} \leq r$$
$$\Rightarrow \frac{\bar{\beta}}{\beta} \leq 2^{r/\lambda} \Rightarrow \bar{\beta} \leq 2^{r/\lambda} \cdot \beta$$

Note that in both cases, the function $\Omega(p)$ is positive in $(0, 1)$.

## E  Supplementary Experiments

**Compute resources.** We ran our experiment using Nividia A-100 and H-100 Tensor Core GPUs. For each experiment only one GPU was used. We use the A-100 GPU for the CIFAR-10 dataset under ResNet setup, and the conformal risk control experiment. The rest of the results use H-100 as the compute resource.

**Experimental setup.** For the CIFAR-10 datasets we evaluate the results over 2048 test samples for ResNet model and 10000 images for the ViT models. For the ImageNet since the number of classes are 1000, we report our results over 5000 images for ViT models and 50000 images on ResNet models. Ultimately the number of samples does not influence the empirical results. The number of Monte Carlo samples are initially set to 500 for CIFAR and 300 for ImageNet. For each experiment, and for the reported sample rate we cut the precomputed samples, from the reported number.

Our results are reports over 100 runs (except the conformal risk control which is over one run. In each run we sample the 10% of the points as the calibration set. For conformal risk control we report the result on 300 images where 100 random images from it is taken for the calibration. Ultimately the size of the calibration set does not effect the final performance. As the calibration set gets larger the distribution of the coverage probability concentrates around $1 - \alpha$.

The time to compute the logits for the CIFAR-10 dataset is 1:30:56 (ViT with 10000 datapoints and 500 samples), and for the ImageNet dataset it is 13:52:11 (ViT with 5000 datapoints and 300 MC samples). For the ImageNet, and the ResNet model this number is 2:52:11 (for 50000 datapoints and 1000 samples).

Table 4: Empirical coverage and average set size for different radii ($r$), for CIFAR-10 dataset with ResNet model and $\sigma = 0.25$.

| $r$ | Empirical Coverage | Avg Set Size |
|---|---|---|
| 0.06 | $0.936 \pm 0.018$ | $2.156 \pm 0.241$ |
| 0.12 | $0.961 \pm 0.014$ | $2.646 \pm 0.306$ |
| 0.18 | $0.981 \pm 0.010$ | $3.315 \pm 0.478$ |
| 0.25 | $0.990 \pm 0.008$ | $4.178 \pm 0.798$ |
| 0.37 | $1.000 \pm 0.000$ | $10.000 \pm 0.000$ |
| 0.50 | $1.000 \pm 0.000$ | $10.000 \pm 0.000$ |
| 0.75 | $1.000 \pm 0.000$ | $10.000 \pm 0.000$ |

Table 5: Empirical coverage and average set size for different radii ($r$), for CIFAR-10 dataset with ResNet model and $\sigma = 0.5$.

| $r$ | Empirical Coverage | Avg Set Size |
|---|---|---|
| 0.06 | $0.921 \pm 0.020$ | $2.684 \pm 0.244$ |
| 0.12 | $0.937 \pm 0.018$ | $2.937 \pm 0.285$ |
| 0.18 | $0.951 \pm 0.016$ | $3.236 \pm 0.356$ |
| 0.25 | $0.966 \pm 0.014$ | $3.741 \pm 0.530$ |
| 0.37 | $0.980 \pm 0.010$ | $4.500 \pm 0.560$ |
| 0.50 | $0.990 \pm 0.007$ | $5.300 \pm 0.712$ |
| 0.75 | $1.000 \pm 0.000$ | $10.000 \pm 0.000$ |

[Left] $\sigma = 0.25$

| | 0.06 | 0.12 | 0.18 | 0.25 | 0.37 | 0.5 | 0.75 |
|---|---|---|---|---|---|---|---|
| 30 | 7.8 | 7.3 | 6.7 | 5.9 | 0 | 0 | 0 |
| 40 | 1.3 | 7.3 | 6.7 | 5.8 | 0 | 0 | 0 |
| 50 | 0.87 | 1 | 6.7 | 5.8 | 0 | 0 | 0 |
| 60 | 0.73 | 0.65 | 0.38 | 5.8 | 0 | 0 | 0 |
| 70 | 0.57 | 0.39 | 0.083 | -0.021 | 0 | 0 | 0 |
| 80 | 0.39 | 0.2 | -0.13 | -0.34 | 0 | 0 | 0 |
| 90 | 0.36 | 0.16 | -0.18 | -0.48 | 0 | 0 | 0 |
| 100 | 0.31 | 0.051 | -0.37 | -0.71 | 0 | 0 | 0 |
| 150 | 0.067 | -0.17 | -0.62 | -1.1 | -6.1 | 0 | 0 |
| 200 | -0.016 | -0.33 | -0.7 | -1.3 | -6.4 | 0 | 0 |

$r$

[Right] $\sigma = 0.5$

| | 0.06 | 0.12 | 0.18 | 0.25 | 0.37 | 0.5 | 0.75 |
|---|---|---|---|---|---|---|---|
| 30 | 7.3 | 7.1 | 6.8 | 6.2 | 5.5 | 4.6 | 0 |
| 40 | 2.3 | 2.4 | 2.7 | 6.2 | 5.6 | 4.7 | 0 |
| 50 | 1.8 | 1.7 | 1.9 | 1.8 | 5.6 | 4.6 | 0 |
| 60 | 1.4 | 1.5 | 1.4 | 1.3 | 1.2 | 4.7 | 0 |
| 70 | 1.2 | 1.3 | 1.1 | 0.93 | 0.82 | 1.1 | 0 |
| 80 | 1.1 | 0.95 | 1 | 0.67 | 0.55 | 0.51 | 0 |
| 90 | 0.95 | 0.86 | 0.79 | 0.54 | 0.26 | 0.39 | 0 |
| 100 | 0.88 | 0.77 | 0.63 | 0.43 | 0.055 | -0.16 | 0 |
| 150 | 0.56 | 0.46 | 0.32 | 0.084 | -0.31 | -0.74 | -4.1 |
| 200 | 0.38 | 0.29 | 0.071 | -0.26 | -0.55 | -0.86 | -4.6 |

$r$

Figure 11: Comparison of BinCP and RCP1 in terms of $|\mathcal{C}_{r,\mathrm{BinCP}}| - |\mathcal{C}_{r,\mathrm{RCP1}}|$ (higher (green) shows better performance for RCP1) across various radii and sample rates. Results are on the ResNet model and for the CIFAR-10 dataset. [Left] $\sigma = 0.25$, and [Right] $\sigma = 0.5$. Note that the numbers are in terms of difference and to compute the absolute number Table 4, and Table 5 can be used as reference.

**Set size experiment.** Tables Table 4 and Table 5 report the empirical coverage and average prediction set size of RCP1 for different radii $r$ on the CIFAR-10 dataset using a ResNet model under two noise levels, $\sigma = 0.25$ and $\sigma = 0.5$, respectively. We also report the result of the ImageNet dataset (for the ResNet model) in Table 6. Specifically for this dataset, because of the large number of classes we also reported the proportion of the sets below specific sizes (1, 3, 5, and 10). As expected, increasing the radius $r$ results in a more conservative setup and hence higher coverage on the clean points. For CIFAR-10 dataset Fig. 11, and for the ImageNet dataset Fig. 12 visualize the comparative performance between BinCP and RCP1 across various radii and sampling budgets. These results are on the ResNet model. The heatmaps show the difference in set sizes $|\mathcal{C}_{r,\mathrm{BinCP}}| - |\mathcal{C}_{r,\mathrm{RCP1}}|$, where positive (green) values indicate that RCP1 provides smaller or more efficient sets. RCP1 generally outperforms BinCP across low sample rates, especially for smaller radii and moderate sampling budgets. The reference tables (Tables Table 4 and Table 5, Table 6) can be used to interpret these differences in absolute terms.

**Proportion of small sets.** As also discussed in § 4 although the proportion of sets with size less than a threshold shows how applicable a CP algorithm is, it can be misleading – a CP framework can return many false prediction sets with very small set size. Therefore alongside the proportion of these sets we should also report their coverage. We show these results in Fig. 13. Our observation is that all setups result in sets with coverage higher than the determined level. Note that in terms of proportion RCP1 stands somewhere between BinCP with 64 and 128 samples which aligns with our aforementioned intuition.

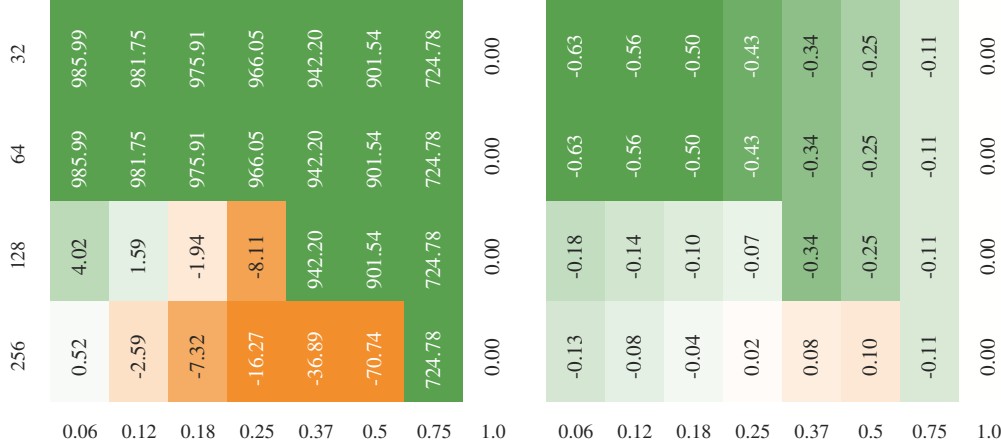

Figure 12: [Left] Comparison of the average set size $|\mathcal{C}_{r,\mathrm{BinCP}}| - |\mathcal{C}_{r,\mathrm{RCP1}}|$ and [Right] the proportion of the sets with size $\leq 10$ expressed in BinCP - RCP1. In both plots green shows that RCP1 is performing better. To convert the relative difference to absolute number Table 6 can be used as the reference.

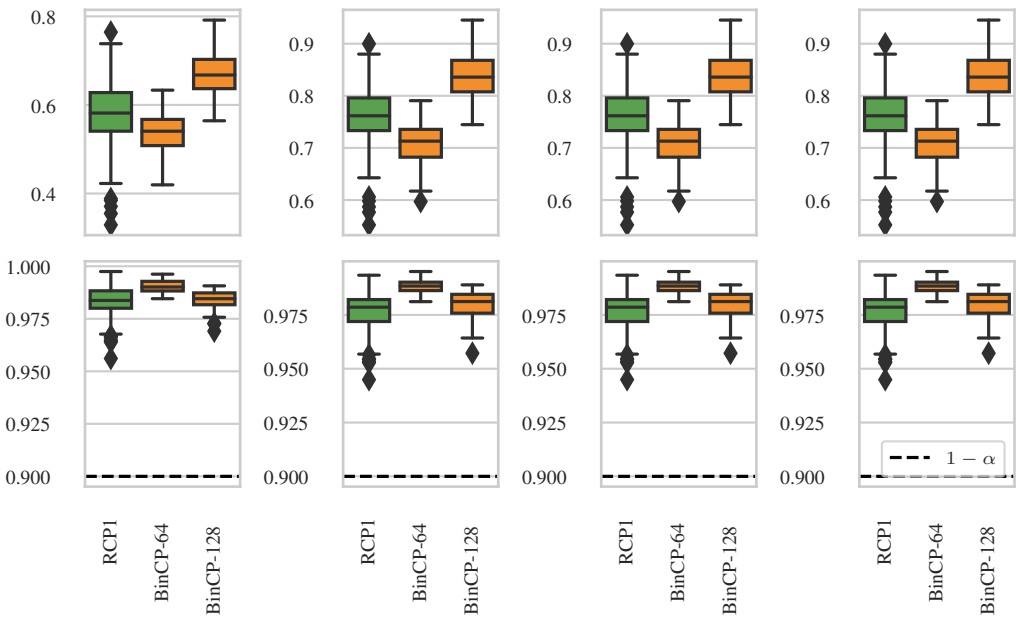

Figure 13: [Up] The proportion and [Bottom] the coverage of the prediction sets with size [From left to right] $|\mathcal{C}| \leq 1$, $|\mathcal{C}| \leq 3$, $|\mathcal{C}| \leq 5$, $|\mathcal{C}| \leq 10$.

Table 6: Statistics from RCP1 across various radii. The results are for ImageNet dataset and the ResNet model.

| $r$ | Avg Set Size | Emp. Coverage | $\mathcal{C} \leq 1$ | $\mathcal{C} \leq 3$ | $\mathcal{C} \leq 5$ | $\mathcal{C} \leq 10$ |
|---|---|---|---|---|---|---|
| 0.06 | $14.013 \pm 1.787$ | $0.921 \pm 0.008$ | $0.139 \pm 0.010$ | $0.311 \pm 0.018$ | $0.430 \pm 0.025$ | $0.626 \pm 0.035$ |
| 0.12 | $18.246 \pm 2.606$ | $0.936 \pm 0.009$ | $0.120 \pm 0.010$ | $0.274 \pm 0.019$ | $0.383 \pm 0.024$ | $0.560 \pm 0.032$ |
| 0.18 | $24.095 \pm 3.645$ | $0.951 \pm 0.008$ | $0.101 \pm 0.010$ | $0.239 \pm 0.018$ | $0.336 \pm 0.023$ | $0.501 \pm 0.031$ |
| 0.25 | $33.953 \pm 5.744$ | $0.964 \pm 0.007$ | $0.082 \pm 0.008$ | $0.201 \pm 0.017$ | $0.288 \pm 0.022$ | $0.432 \pm 0.033$ |
| 0.37 | $57.802 \pm 10.753$ | $0.979 \pm 0.005$ | $0.058 \pm 0.008$ | $0.151 \pm 0.017$ | $0.219 \pm 0.022$ | $0.337 \pm 0.031$ |
| 0.50 | $98.464 \pm 19.136$ | $0.989 \pm 0.003$ | $0.036 \pm 0.006$ | $0.104 \pm 0.016$ | $0.160 \pm 0.021$ | $0.252 \pm 0.029$ |
| 0.75 | $275.222 \pm 88.968$ | $0.998 \pm 0.002$ | $0.012 \pm 0.006$ | $0.036 \pm 0.016$ | $0.063 \pm 0.025$ | $0.115 \pm 0.039$ |
| 1.00 | $1000.000 \pm 0.000$ | $1.000 \pm 0.000$ | $0.000 \pm 0.000$ | $0.000 \pm 0.000$ | $0.000 \pm 0.000$ | $0.000 \pm 0.000$ |

Table 7: RCP1 for conformal regression. We report the empirical coverage and the interval length across varius radii.

| $r$ | Empirical Coverage | Interval Width |
|---|---|---|
| 0.00 | $0.900 \pm 0.005$ | $0.371 \pm 0.012$ |
| 0.12 | $0.920 \pm 0.005$ | $0.426 \pm 0.014$ |
| 0.25 | $0.938 \pm 0.004$ | $0.494 \pm 0.018$ |
| 0.50 | $0.963 \pm 0.003$ | $0.667 \pm 0.026$ |

**Regression Experiment**  For robust conformal regression with RCP1 we use the Udacity and originates from Nvidia's DAVE-2 system[4]. The input of this task is an scene, and the task is to estimate the steering angle of the car. The output range is from -1 (completely steering right) to 1 (left). For this task we finetune a ResNet18 model [15] on images augmented with isotopic Gaussian noise with $\sigma = 0.5$. We run finetuning for 200 epochs. We use the same $\sigma$ for augmenting the input in RCP1. We set $1 - \alpha = 0.9$, and evalute on $r \in \{0.12, 0.25, 0.5\}$. To the best of our knowledge our result is the first robust conformal prediction with randomized smoothing for regression task. Table 7 compares the interval length and empirical coverage across various radii.

