# OpenReview forum: "One Sample is Enough to Make Conformal Prediction Robust"
_NeurIPS.cc/2025/Conference — NeurIPS 2025 poster_

### Official Review · Reviewer_vsiq · 2025-06-29

**Clarity:** 2
**Significance:** 3
**Originality:** 3
**Rating:** 5
**Confidence:** 4

**Summary:**

The paper tackles the high computational cost of current smoothing‐based robust conformal prediction (RCP) methods and shows that in fact a single noise‐augmented inference suffices. Their key insight is to “certify the conformal procedure itself” rather than individual model scores: by applying any binary robustness certificate (e.g. Gaussian or Laplace smoothing bounds for ℓ₂ or ℓ₁ threats) directly to the coverage probability of the vanilla conformal predictor, one can adjust the calibration quantile to restore a 1–α worst‐case coverage guarantee. This yields **RCP1**, a black‐box, score‐agnostic method that needs only one perturbed forward pass per test point.

**Contributions**

1. **Method**: Introduce RCP1, which uses a single randomly perturbed sample plus an off‐the‐shelf binary certificate to attain robust coverage, eliminating the need for costly Monte Carlo sampling.
2. **Generality**: Works with any model, any conformity score (classification or regression), and any smoothing/distributional threat model (Gaussian, Laplace, uniform; ℓ₂, ℓ₁, etc.).
3. **Extension**: Show how to convert smoothing‐based RCP1 into a robust *conformal risk control* procedure, demonstrated on semantic segmentation (Cityscapes) to control false‐negative rate.
4. **Empirical validation**: On CIFAR-10 and ImageNet with both ResNets and ViTs, RCP1 matches the set‐size vs. coverage trade-off of state-of-the-art methods like BinCP (which use 64–128 samples) while being orders of magnitude faster (one vs. ∼100 inferences) (Fig. 1-middle) and achieving the same 1–α robust coverage under adversarial attacks (Fig. 1-left).

**Questions:**

- typo in the equation below line 121
- Another slight clarity issue is that the paper introduces many acronyms (CP, RCP, RCP1, BinCP, CAS, etc.) and shorthand notations for different methods. While it does define them (often by citing the relevant references), a concise table or summary of methods might have helped the reader keep track. That said, the authors do consistently remind the reader of what each baseline is (e.g. noting that BinCP uses p-quantiles or that CAS uses CDF information), which mitigates confusion.
- The writing could have been further improved with a bit more exposition on the certificate calculations and perhaps by explicitly contrasting RCP1’s procedure with standard conformal prediction in a step-by-step manner. These are minor suggestions; the core ideas and results are still clearly conveyed.
- The method is described as agnostic to the setup (classification vs. regression), but all experiments are on image classification. It would be valuable for the authors to discuss or demonstrate the applicability of RCP1 in other settings – for example, conformal prediction for regression tasks or other data modalities. Even a brief discussion or a toy experiment (perhaps in the appendix or supplemental material) showing how robust conformal prediction works for a regression problem would strengthen the paper’s claim of generality
- The proposed RCP1 method relies on the availability of a “binary certification” for model predictions (for example, the paper uses randomized smoothing to certify robustness of a classifier). A question arises about how practical and general this requirement is. The authors should clarify to what extent RCP1 depends on a specific certification technique and whether any robust model certificate can be used interchangeably

**Ethical Concerns:**

["NO or VERY MINOR ethics concerns only"]

**Limitations:**

Yes

**Paper Formatting Concerns:**

Yes

**Quality:**

3

**Strengths And Weaknesses:**

Strengths:
- This paper proposes RCP1, a novel robust conformal prediction method that requires only a single model inference per input, yet still guarantees the desired coverage under worst-case perturbations. The theoretical contribution is solid – the authors prove that by adding one instance of random noise to each example (both in calibration and at test time) and leveraging a robustness certificate, the conformal coverage guarantee can be maintained even for an adversarially perturbed input. In particular, they choose a slightly more conservative nominal coverage level $1-\alpha'$ such that the certified lower bound on coverage under perturbations is at least $1-\alpha$. This key insight – “certify the conformal procedure itself rather than individual scores” – is innovative and allows RCP1 to inherit some robustness from even a single noise-augmented forward pass.
- The paper provides formal propositions (e.g. Proposition 1) and defers detailed proofs to the appendix, indicating a rigorous theoretical underpinning. All assumptions and theoretical results appear to be clearly stated (per the checklist), lending credibility to the claims (thought I did not have time to check all the details - I have read in detail the proof of proposition 1).
- Empirically, the evaluation is thorough and convincing. The authors test RCP1 on multiple datasets and models, including CIFAR-10 and the large-scale ImageNet with both ResNet and Vision Transformer backbones, as well as a challenging semantic segmentation task for conformal risk control. The experiments demonstrate that RCP1 achieves robust prediction sets of comparable size to state-of-the-art multi-sample methods while using dramatically less computation. For example, on ImageNet, RCP1’s average prediction set size is similar to that of the prior BinCP method when BinCP uses about 64–128 Monte Carlo samples per input. This is a remarkable improvement in efficiency – RCP1 attains essentially the same set size and coverage as BinCP with ~100 forward passes, using only one forward pass. The paper quantifies computational gains: for instance, processing 5000 ImageNet images with a ViT model takes ~2.8 minutes with RCP1 versus ~5.9 hours with an equivalent BinCP (128-sample) approach – a huge speedup that underscores the practical value of the method.
Weaknesses:
- One concern is that RCP1’s reliance on a single random draw per input introduces higher variance in the prediction sets. The authors acknowledge this limitation: “RCP1 shows considerably more randomness in the prediction sets compared to BinCP”, due to the inherently random one-sample procedure. In other words, the exact set of labels returned by RCP1 can vary across different runs (or different noise instantiations) on the same input, more so than methods that effectively average over many samples. While the coverage probability is still guaranteed in expectation, this increased variability might be undesirable in certain applications that demand deterministic or highly stable predictions.
- the paper does cover a lot of technical ground, which can make certain portions dense. A reader not already acquainted with randomized smoothing certificates might find it challenging to parse the certificate optimization (Equation 4) and related discussion without consulting the appendix. The authors rely on references (e.g. to Yang et al. [22]) for some of the certificate details, which is appropriate, but it means that part of the argument is somewhat terse in the main text. For example, the statement “the optimization in Eq. 4 is translation invariant... we can use a fixed set of canonical directions” hints at non-trivial results from prior work that are not fully explained. Similarly, Section 3.1 briefly mentions how to compute certified probabilities $c_{\downarrow}[p,B]$ for various smoothing schemes by “small changes” to Yang et al.’s recipe, which could be hard to grasp without familiarity with [22].
- While the idea is novel, it builds heavily on prior work, and in some respects could be seen as an incremental improvement. The paper itself references BinCP (Zargarbashi & Bojchevski, ICLR 2025) as a state-of-the-art robust CP method that introduced the concept of using a “single binary certificate” to reduce sample complexity. In fact, the authors of this submission appear to be extending their own previous work (BinCP) – the terminology and some methodology overlap (e.g. use of Clopper-Pearson confidence intervals, references [23] and [24] which are prior methods by presumably the same group). From that standpoint, RCP1’s novelty is mainly in eliminating the remaining Monte Carlo sampling that BinCP still required.
- -A final point is that the method’s correctness assumes access to a reliable robustness certificate for the base model (e.g. the ability to lower-bound the probability $P(f(x+\epsilon)$ predicts class $y$) for all $\epsilon$ in a perturbation ball). The approach is only as good as these certificates – if a model cannot be certified for a meaningful radius (or if the certificate probability $c_{\downarrow}$ is too low), the resulting prediction sets could become trivially large.

---

> ### Author Rebuttal · Authors · 2025-07-31
>
> We thank the reviewer for reading our paper and we are happy that the reviewer liked it.
>
> 1. Thank you for pointing out the typos, surely we would modify it in the camera ready version.
> 2. The reviewer is right. We should definitely add a section on a summary of how each prior method is functioning. We would definitely add that to the camera ready. We also agree that the paper is introducing many term and notations which makes it uneasy to read.
> 3. Of course; we will add a section in the appendix with a brief discussion on how robustness certificates, and how CP work. Hopefully that would make the reading of the paper easier.
> 4. Sure. We have discussed its adaptability to regression while showing something like that is definitely helpful in illustrating it. By the time of publication we did not find a good example of smoothing based regression on a large data. But for the camera ready we would try to add such experiment at least with a synthetic dataset. We also like to draw your attention to the image segmentation certificate in Figure 7 which uses single-sample robust conformal risk control.
> 5. Indeed any smoothing-based certificate is directly applicable to both RCP1, and BinCP for free. If the reviewer refers to the more practical smoothing noises, we can use the recipe from [22] in combination prop 2. If the question is whether RCP1 adapts with other types of robustness guarantees including verifiers, it is not adaptable. But randomized smoothing shows to be very strong in terms of certifiable radii, and in our case efficiency of the set size. And since this method is model (and score) agnostic in can easily adapt to any machine learning model.

---

### Official Review · Reviewer_p6ny · 2025-07-03

**Clarity:** 2
**Significance:** 2
**Originality:** 2
**Rating:** 4
**Confidence:** 2

**Summary:**

The paper introduces RCP1, a robust conformal prediction method that overcomes the computational hurdles present in smoothing-based RCP approaches. By leveraging randomized smoothing, the proposed method achieves robustness against adversarial perturbations using only a single forward pass per input. The experiments are conducted on CIFAR-10 and ImageNet for classification and cityscapes for risk control settings.

**Questions:**

See above.

**Ethical Concerns:**

["NO or VERY MINOR ethics concerns only"]

**Final Justification:**

The reviewer is not fully convinced by the paper's technical assumptions as well as its theoretical novelty; also, some notations are hard to follow.

**Quality:**

2

**Strengths And Weaknesses:**

Strengths

- The paper aims to address the computational bottleneck in existing smoothing-based robust conformal prediction methods, which is well-motivated.

- The integration of noise-augmented inference with robust conformal prediction offers an easy-to-implement solution.

- The paper is well written.

Weaknesses

- The robustness guarantee in RCP1 is marginal over the randomness introduced by the smoothing process. Coverage is only guaranteed on average over the added noise, rather than under worst-case scenarios as in prior work. Specifically, the guarantee depends on the type of added noise (e.g., Gaussian perturbations), and does not hold uniformly across all inputs or perturbations. This also weakens the comparison with baselines that aim to certify worst-case robustness, potentially leading to unfair or misleading results.

- The authors primarily focus on Gaussian smoothing, which limits the practicality of the approach, as real-world noise settings are often more complex and structured.

- The study heavily builds upon prior work, making its theoretical contribution insufficiently novel.

---

> ### Author Rebuttal · Authors · 2025-07-31
>
> We thank the reviewer for reading our paper and for the comments. Here we discuss the raised weaknesses:
>
> 1. The are two terms that should be distinguished: (1) the guarantee holds marginal over augmented noise. (2) The guarantee (marginal over noise) holds for the **worst case perturbation**. So here the noise is what the defender adds to any (including perturbed) input, the perturbation is what the adversary introduces to the clean data. Our guarantee holds marginal to the noise we introduce not the perturbations the adversary applies. This is a very important distinction of a certified worst-case robustness (ours) and an average-case robustness like [12] (in our references). We discussed this further in line 377. Formally for $\tilde{x} = x +  \delta$ we provide a guaranteed prediction set using $\tilde{x} + \epsilon = x + \delta + \epsilon$ which is marginal over $\epsilon$ for the worst case $\delta$. This means that our comparison with BinCP[24] is fair as we both target the same worst-case guarantee. Moreover, being marginal over random noise is not uncommon in the literature. For example the highly popular APS score function introduces uniform random noise, and there the 1-alpha guarantee is also only marginal over the introduced noise. See Chapter 9.1 in [add reference].
>
> 1.  We draw the reviewer’s attention to Fig 2-right and Fig 5-right where we discuss noises other than isotropic Gaussian, specifically Laplace and Uniform noise. Indeed Section 3.1 modifies the result from Yang et al [22] to adapt RCP1 with smoothing of all shapes and sizes. RCP1 itself is independent to the noise type — any smoothing scheme, and threat model can be adapted to RCP1 for free.
> 2. **In terms of result:** Existing worst-case CP guarantees are categorized into two main regimes: (1) smoothing based approaches which are robust for large radii while requiring may MC samples (computationally expensive), and (2) verification and Lipschitz based approaches which are efficient but only support very small radii. Our work is computationally efficient and works for large radii which is the first method which has both benefits.
>
>     **In terms of theoretical novelty:** In order not to use MC sampling, we need to rely on the non-exact (probabilistic) setup which is unique to the case of conformal prediction — we can not apply the same method to have a certified top-class prediction. Similarly other methods did not need to prove that the certificate is convex. Notably, the proof that allows us to work with a single sample is completely orthogonal to the other smoothing based CP approaches, as they redefine the score function and show the score changes are bounded; that is basically why the need to estimate the score by MC sampling. Here we prove it by leveraging and lower bounding the coverage probability which is not needed to be estimated.

---

### Official Review · Reviewer_cNni · 2025-07-03

**Clarity:** 1
**Significance:** 2
**Originality:** 2
**Rating:** 4
**Confidence:** 3

**Summary:**

This paper uses the binary certificate technique to develop a single sample robust CP (RCP1), which reduces model forward passes in constructing the prediction set. The experiments show that RCP1 achieves robust prediction sets with comparable sizes to state-of-the-art methods while reducing computational overhead.

**Questions:**

1. In Line 41, what does "vanilla CP combined with noise-augmented inference" mean?

2. What is the proposed prediction set and coverage guarantee?

3. How can you make sure the lower bound in Proposition 1 is greater than $1-\alpha$? If this lower bound is large, how can we say "conformal prediction combined with single randomized augmented inference already shows robustness"?

4. What is the form of $\mathcal{C}_r$ in Line 108?

**Ethical Concerns:**

["NO or VERY MINOR ethics concerns only"]

**Final Justification:**

The authors have clarified that the error can be fixed.

**Limitations:**

yes

**Quality:**

1

**Strengths And Weaknesses:**

**Strengths**

This paper introduces a new technique for robust CP under adversarial noise.

**Weaknesses**

**1. Notations are heavy, confusing, and not easy to follow.**

For example, from Lines 115 to 121, there are multiple definitions for $q$. In addition, Lines 80-85 are a recap of previous work [24], and the notations are not necessary here. Moreover, some notations are not explained here, such as $c$ with arrows.

**2. The writing is not clear.**

The authors should provide a clean algorithm or definition for the proposed prediction set. After the first read, I didn't know what the final prediction set is, not to mention the implementation.

**3. Potential theoretical error.**

In Line 119, if $q$ is defined in Proposition 1, $P(S_{n+1} > q | E_+) \ge 1-\alpha$ seems wrong because given $E_1,...,E_{n+1}$, the quantile $q$ is not symmetric to $(X_1,Y_1),...,(X_{n+1},Y_{n+1})$.

---

> ### Author Rebuttal · Authors · 2025-07-31
>
> We thank the reviewer for the constructive feedback. First we respond to the technical flaw, then we address other raised concerns and questions.
>
> **Potential Theoretical Error.** Thanks for pointing that out. Turns out the reviewer is right about the flaw in line 119, however since the conclusion is about coverage probability marginal over noise and data, the rest of the proof holds independently of this statement.
>
> *First - Example of why the reviewer is right:* Assume a one dimensional data x randomly in range of [0, 1], and the noise to be also uniform random from the same range. And the score to be S = X + E. Then one set of fixed $E_i$’s is $E_1 = 0, E_2 = 0.1, \dots, E_{10} = 1$. From the definition of the score function the $1 - \alpha$ quantile is the addition of the two quantiles over the X’s and E’s. While the quantile over X’s holds the same, the one over E’s introduces a bias since the noises are ordered, leaving the overall quantile inequality invalid.
>
> *Importantly the proof remains valid even without this argument.* This is intuitive assuming that the noises augmentation is a part of the model inference. Moreover, for other randomized score functions like APS the guarantee similarly holds over the marginalization of the noise. We refer to Chapter 9.1 of [2] showing the same argument applies for any randomised score function — a random noise augmentation of the input is essentially a randomised score function. In short, the 1 - alpha guarantee holds because noise-augmentation with i.i.d. noise is symmetric. The rest of the proof does not rely on the second half of the line 119. Thank you for pointing that out. In the next version, we’ll remove this part and the other occurrences of the same argument which is line 129 and the footnote in page 4.
>
> **Heavy and Hard-to-Understand Notations.** In light of the reviewer’s comment we found some typos that made it harder to read:
>
> - The first half of line 119: The equation is $\Pr[\hat{S}_{n+1} \ge \hat{q}] \ge 1 - \alpha$ for $\hat{q} = ...$.
> - The first half of line 83: The equation is $\mathcal{C} _{\mathcal{B}}(\boldsymbol{x} _{n+1}) = \{y:s(\tilde{\boldsymbol{x}} _{n+1}, y) \ge \bar{q}\}$.
>
> Also regarding the notations $c^\uparrow$, and $c^\downarrow$, we adapted them from [24] and defined in line 81 as certified upperbounds and lowerbounds of the score, although we agree that it is necessary to bring the rigorous notation which says: $c^\downarrow[f, x, \mathcal{B}]$ is the certified lowerbound for function $f$ in $\mathcal{B}(x)$. We’ll surely add modifications in the camera-ready version. If the reviewer has a concrete suggestion on how to improve the notation we would gladly consider it. We also try to reduce the quantity of the variables in a way that doesn’t affect the mathematical rigor.
>
> **Unclear Writing and Algorithm.** We agree that an algorithm would enhance the understandability of the paper. We ask the reviewers attention to the lines 51 to 54 where we discuss the high level implementation. In general the algorithm is exactly same as conventional CP, only with applying noise before computing the score functions and finding the upper bound $1 - \alpha’$. We will add the following peudocode to the camera ready paper.
>
> **Algorithm 1: RCP1 – Single Sample Robust Conformal Prediction**
>
> ---
>
> - **Input:**
>
> The calibration set
> $\mathcal{D} _{\mathrm{cal}} = \{(x _i, y _i)\} _{i=1}^ n$
>
> The score function
> $s: \mathcal{X} \times \mathcal{Y} \to \mathbb{R}$
>
> The test point
>  $x_{n+1}$,
>
> target coverage level $1 - \alpha$
>
> perturbation ball $\mathcal{B}_r$
>
> smoothing scheme $\xi$
>
> - **Output:**
>   $$\mathcal{C} _r(x _{n+1}) \subseteq \mathcal{Y}$$
>
> 1. **Step 1.** Draw noise from the smoothing scheme, compute conformity scores for augmented calibration data:
>    ```text
>    for i = 1 to n do
>        \epsilon_i ∼ \xi
>        s_i \gets s(x_i + \epsilon_i, y_i)
>    end for
>    ```
>
> 2. **Step 2.**
>    - Compute corrected level:
>      $$1 - \alpha' = c^{\uparrow}[1 - \alpha, \mathcal{B}_r]$$
>    - Compute the $$ (1 - \alpha') $$ quantile with correction:
>      $$q \leftarrow Q(\alpha'; \{s_1, \dots, s_n\})$$
>
> 3. **Step 3.** Construct the prediction set for the test input:
>    $$\mathcal{C} _r(x _{n+1}) = \{y \in \mathcal{Y} : s(x _{n+1} + \epsilon _{n+1}, y) \ge q\}$$
>
> 4. **Return** $$\mathcal{C} _r(x _{n+1})$$
>
> ---
>
> **Questions:**
>
> 1. It means if you do not apply any robustness guarantee, and use vanilla CP, only if instead of inference over $x$, we forward the model with $x + \epsilon$, the resulting CP shows some robustness. This robustness means that the amount by which the empirical coverage deviates from the guarantee is significantly less than the case where we only apply the forward on $x$. As you see in Figure 1 - left, the dashed lines (which are CP without changing the quantile level) do not drastically drop as the red line drops.
> 2. The proposed prediction set for the potentially perturbed $\tilde{x}$ is $\mathcal{C} _r(x _{n+1}) = \{y \in \mathcal{Y} : s(x _{n+1} + \epsilon _{n+1}, y) \ge q\}$, where   $q \leftarrow Q(\alpha'; \{s_1, \dots, s_n\})$ with  $1 - \alpha' = c^{\uparrow}[1 - \alpha, \mathcal{B}_r]$. And here the guarantee is the same $1 - \alpha$ for any worst-case adversarial perturbation up to radius $r$.
> 3. The lower-bound in Prop 1 is not greater than $1 - \alpha$. It is lower, but using the randomized smoothing (and in our case noise augmentation) helps us for this lower-bound not to get too deviated. For further clarification: we show that vanilla CP over noise augmented inference leads to a coverage guarantee $c^\downarrow[1-\alpha, \mathcal{B}]$, which means that by calibrating to $c^\uparrow[1-\alpha, \mathcal{B}]$ allows us to achieve $1 - \alpha$ coverage. This is via lemma 3 in [24].
> 4. It can be of any form, including the definition from [24] or our definition. The argument in line 108 is a generalised version of this argument: “if the clean point $x$ is covered by vanilla CP not accounting for perturbation radius $r$, then robust CP also covers the perturbed point $\tilde{x}$. Notably there, the argument is 0/1 here we discuss that the argument should say that the robust prediction set $C_r$ has a higher probability to cover the perturbed point compared to the vanilla set $C_0$ for the clean point.
>
> Once again we thank the reviewer for pointing out the mistake, we surely remove it from the proof, but nevertheless the proof holds even without that statement.

---

> > ### Comment · Reviewer_cNni · 2025-08-02
> >
> > I appreciate the authors' responses, which have largely addressed my concern, and I will increase my rating.

---

### Official Review · Reviewer_MZPH · 2025-07-03

**Clarity:** 4
**Significance:** 2
**Originality:** 3
**Rating:** 5
**Confidence:** 3

**Summary:**

The paper studies robust conformal prediction (RCP) by randomized smoothing. The authors show that with a single forward pass on randomly perturbed input can already attain robustness. Interestingly, the empirical findings support that RCP1 produces smaller set sizes compared to BinCP when sample rates are small. Additionally, RCP1 is agnostic to the distribution of inputs and scores.

**Questions:**

1.	Is the computation of the certified lower bound for RCP1 the main difficulty comparing to methods such as BinCP? If so, is the limitation on the choice of smoothing methods or assumptions?
2.	It is interesting to see when sample rate is low, RCP1 consistently produces smaller prediction size. Is it purely a consequence of randomness in sampling for BinCP by Monte Carlo? Can the authors provide more insights, or whether it is possible to improve prediction set size when sample size is moderate (e.g., between 1 to 100).

**Ethical Concerns:**

["NO or VERY MINOR ethics concerns only"]

**Final Justification:**

The authors clarified some of my initial confusion. The work is interesting and requires some final touch-ups for the presentation and clarity. I remain my recommendation for the paper.

**Limitations:**

See "Strengths and Weakness".

**Paper Formatting Concerns:**

No concerns noted.

**Quality:**

3

**Strengths And Weaknesses:**

Strengths:

1.	The paper is well-written, clearly organized and easy to follow.
2.	The theorems are well presented and are sound for the parts that I checked.
3.	The experiments indeed show favorable properties of RCP1 especially in terms of computational time, and comparable quality when sample size is small. In particular, the empirical findings support that RCP1 generate smaller prediction set size when sample rate is low.

Weakness:

1.	The comparison of RCP1 and previous methods using multiple samples is obscure. For example, a parallel comparison between RCP1 and BinCP can be beneficial.

---

> ### Author Rebuttal · Authors · 2025-07-31
>
> We are very thankful that you read the paper, and very grateful that you liked it.
>
> **Parallel comparison with BinCP**. If by parallel comparison you refer to running both up to the same wall-clock time, indeed BinCP returns full sets if limited to the time RCP1 returns its result. Besides the sampling, BinCP and RCP1 both apply a single binary certificate; which means that although the binary certificate is negligible in computational time, they both have same cost in terms of calling the certificate. The only difference is that BinCP needs more than one forward pass and with only one forward pass (equal or parallel setup) it returns trivial sets of $\mathcal{C}(x) = \mathcal{Y}$.
>
> This also answers the first question. Additionally the bottleneck in BinCP is independent to the choice of the smoothing scheme. It is only due to several runs of the model on the same data. As shown empirically.
>
> **Second question.** We think the reviewer refers to when sample rate is low in BinCP — otherwise what we show is that RCP1 only works with a single sample. The set size for BinCP increases up to returning all possible classes. Here we provide an intuitive reason behind it:
>
> When the score is defined as some statistical term over the model and smooth input, since the certificate works with exact probabilities, we should estimate that statistics. For example consider the case where the score function is defined as the mean of the score distribution (over smooth input). There is a ground truth value, but finding that value is intractable. Instead we compute an interval which covers that true score with a very high probability (e.g. 0.999) — known as confidence interval.  The size of this confidence interval decreases by introducing more MC samples. For correctness of robust CP we should (conservatively) take the lower bound of this interval and that is what contributes to increasing the size of the prediction set.
>
> The main contribution of this work is to find another variable to apply the robustness certificate over — a variable that is not needed to be estimated and that is the coverage probability. What we show is that the expected coverage probability (which is already given despite the number of MC samples) is lower bounded. Prior works showed similar for the score function (or quantile of scores which is a function of scores) and scores should be estimated.

---

> > ### Comment · Reviewer_MZPH · 2025-08-03
> >
> > Thank you for your response. It clarifies my confusion over the difference between RCP1 and BinCP. My original impression was they were more similar and there were some easy interpolation between RCP1 to BinCP when sample size for each data point is m. Now I think this work provides more novel insights.
> >
> > Indeed, I think the presentation could be improved to help better understand the key difference. Perhaps a parallel comparison between RCP1 and BinCP as your responses to Reviewer ABhm and cNni could help.
> >
> > My question is then somewhat still similar--if provided data points with multiple samples $S_{i,m} = s(X_i+E_{i,m}, Y_i)$, is there a similar way to have such guarantees (without estimation)? This is clearly beyond the scope of this paper and I think this would be interesting for future work.

---

> > > ### Author Response · Authors · 2025-08-07
> > >
> > > Thank you for your response. We are really happy to hear that the reviewer finds our work novel.
> > >
> > > Surely we would aim for better readability for the camera ready.
> > >
> > > That is definitely a really interesting question. We thank you for pointing that out. We would surely consider it for the future works.

---

### Official Review · Reviewer_ABhm · 2025-07-06

**Clarity:** 1
**Significance:** 3
**Originality:** 2
**Rating:** 4
**Confidence:** 3

**Summary:**

The paper claims that robust conformal prediction can be achieved by using a single sample for each example in the calibration set.
Robust conformal inference is a method for providing a super-set of classes in a classification task that with high probability
(on average over test points) includes the correct class, even when an adversary is allowed to locally perturb the feature vector.
As with (many) other robust conformal prediction, the key to robustness is adding random noise to the input before applying the score function, thereby negating the specific choice of perturbation. However, this cannot overcome the fact that the (smoothed) calibration set is no longer exchangeable with the (smoothed) perturbed feature vector, but rather only with the (smoothed) original vector. Methods that were proposed to bound this extra decay (or extra radius) used many samples (per calibration point) to estimate the post-smoothing mean score or the post-smoothing quantiles. This paper presents an argument that the multi-sample approach is not necessairy to maintain conformal coverage, and therefore reasonably-sized robust conformal set can be estimated with far fewer forward passes. This can make robust conformal prediction much more affordable even when using large prediction models. The experiments compare the conformal prediction levels under adversarial feature choice, and find that for small samples, the new sets are competitive in terms of size and coverage (though have higher variance).

**Questions:**

See above, there are many things I did not understand upon reading the paper.
I think this at least partly reflects on the clarity of the current writup.
But also I would like to better understand the answer to these questions,
and clarifications, especially if they manifest in the text of the paper as well, could definitely result in the increase of scores.

**Ethical Concerns:**

["NO or VERY MINOR ethics concerns only"]

**Final Justification:**

I think I now understand the main claims of the paper,
helped by the discussion with the authors.
I think the ideas are interesting and worth presenting in the conference.
I have therefore increased my score.

However, the paper has substantial clarity issues,
and these surfaced in our discussion.
Important aspects of the methods and the proofs are hidden within the proofs.
The high-level ideas were almost not discussed:
- How the methods works with one sample (an upper bound on the effect of smoothing)
- What are the downstream effects of using a universal but worse case upper bound
instead of estimating a local statistic as the other methods do.
[There are both positive and negative effects. the positive are in the title; the negative,
in potentially reducing the conditional coverage, increased variance, and potentially increased samples sizes,
are only partially discussed in the limitations and are somtimes hidden).

Furthermore, the writing of proofs and the figures / legends can be improved for clarity.

I hope that if the paper is accepted the authors try to fix the final version so to increase clarity and
to reflect our discussions.

**Limitations:**

I thought the discussion of the added variance in using one sample (discussed in two samples in the limitations) should be expanded;
 more generally,
what do we lose when we only use a single sample?
Does it mean that our conformal coverage is less or more uniform
(i.e. is the conditional coverage more or less similar across samples)?

**Quality:**

2

**Strengths And Weaknesses:**

Strengths:
Significance:
For ML algorithms based on large NN, the number of calls to the prediction function is often budgeted due to time,
compute or even monitory constrains. Therefore identifying a robust-conformal algorithm that can be run with considerably less
function calls per sample with only limited impact on the efficiency (e.g. the adversarial radius) can be very important for adoption of these ideas.

Originality:
I think the argument itself for providing much more efficient (in terms of samples) conformal sets is potentially a step forward for the field.
It is hard for me to assess how much is new in the argument or method due to issues of clarity in which I struggled with the paper.

Weakness:
Clarity / Quality:
I admit that I did not manage to understand the main theoretical argument of the paper.
Namely, what are the method by which they claim to change methods that use multiple samples (to estimate lower bounds on the mean region or the quantiles of the distribution) into a method that uses a single sample.

Here are the specific questions I did not have answers to, after multiple readings of this manuscript:
- What are the main reason that other papers state for using multiple samples to estimate the bound-adjustment for robust Conf-Sets,
and how does this paper elevate or pass over those reasons.
- Does the paper introduce "only" a new claim and proof, or also a new robust-conformal prediction algorithm (inc a new way to estimate the bounds c^{arrow down}).
- Moreover, it is not clear what is the precise *algorithm* or bound that used to compute the Conformal Set and in what specific ways is it different from, say, Zargarbashi and Bojchevski bound or Cohen et al or other previous works.
- Do the claims of success of a single sample based bound refer to the empirical or to the theoretical worst-case coverage (e.g. to which one of the curves in the middle of Figure 2).

More generally, I found it very hard to distinguish the trees from the forest:
For examples, in Prop 1 the papers runs through proof without explaining the ideas behind it,
and uses in the statement of the result and the proof an important concept / notation (the lower certified bound c^{down-arrow}) that is  defined in Eq 4 almost one page below.

[This could be my fault here; I am not an expert in the litrature of robust Conformal sets
I am well read in conformal prediction generally, and have manged to read and mostly understand the main cited papers including Yang et al, Cohen et al, and Zargarbashi and Bojchevski; I failed on this paper. ]

Also, in clarity, it is not helpful that the full explanations of the figures is in the appendix; many of these figures where hard to understand without the longer version of the legend, and I did not know the longer legends are in the appendix until I went there to check.

---

> ### Author Rebuttal · Authors · 2025-07-31
>
> We thank the reviewer for reading our paper; we understand that the writing was difficult to follow; and surely we’ll update the camera ready to enhance the readability. Here we discuss the questions and unclear parts. If the results remains yet unclear, we are more than happy to discuss further:
>
> **RCP1 vs other smoothing-based robust CP approaches.** Consider the robustness certificate (Eq 4) as a black-box that receives a probability (or generally any bounded value) and returns a lower (and upper) bound for that probability. Now among the previous methods (RSCP+, CAS, BinCP) all applied this bound on the score function. RSCP+ defines the new score as the mean of the smooth score distribution, CAS does the same only through the CDF of that distribution finds a better bound (a better black-box), and BinCP applies the lower bound on the quantile of the scores. This work derives a lower bound on the coverage probability itself. Then it reverse engineers a higher $1 - \alpha’$ such that its lower bound is above $1 - \alpha$.
>
> *What is different?*  All three prior methods are bounding the score function. Since the robustness certificate works with exact probabilities, we need to estimate that value from Monte Carlo examples (estimate the mean of some smooth scores from m samples), and since we are estimating with finite samples we should account for final sample correction. This correction creates a trade-off: more samples per point results in tighter confidence interval (smaller sets), so we should either get large sets, or run many forward passes.
>
> Oppositely, since we apply the lower bound on a given probability $1 - \alpha$, we do not need to estimate it. Therefore, with only one (noise augmented) inference, we can compute the robust prediction sets.
>
> *What are the challenges?* The proof is generally different, and since this $1 - \alpha$ is the expectation of a Beta distribution itself, we should prove that the smoothing-based lower bound can move from each run to the total expectation. For that we needed to prove that robustness certificates are convex.
>
> Also from the description, we answered that we do not propose a new robustness certificate. We change the way we are using it in conformal prediction to improve the computational efficiency. Same applies to all three mentioned prior works.
>
> *Why we need sampling in prior works?* Since in prior works the object which we wanted to bound within the threat model is the mean (quantile, or any other statistics) of a distribution which we don’t know. And computing that statistics exactly is intractable. Therefore, we need to estimate it via samples, and more samples result in a more precise estimation of that value.
>
> **Figure 2.**
> The solid line is the theoretical lower bound, derived in Proposition 1, which should be (and is) lower than the empirical line which is the dashed line (both are for single sample). So the empirical coverage should be higher than what we compute as a lower bound. The figure shows that indeed the lower bound holds. Note, that this solid line is computed for a target of 1-alpha = 90% — lower bounding the worst-case coverage if it is calibrated with 90%. For example, for r=0.2 we get worst-case coverage of ~81%. If we want worst-case coverage of 90%, as a defense to the perturbations, we find an initial higher 1-alpha=95% which after the lower bound becomes 90%.
>
> **Algorithm.**
> You are right. Although we discuss the general algorithm in the introduction (lines 81-83), still nowhere else we explicitly define the algorithm. In following we bring the pseudocode, and later we will add it to the camera ready version. The algorithm is to do both calibration and defining prediction set same as vanilla CP with two differences: (1) instead of inference over $x$, we should apply inference over one sample from $x + \epsilon$. (2) Instead of $1 - \alpha$ we should aim for coverage guarantee of $c^\uparrow[1- \alpha, \mathcal{b}]$ to attain $1 - \alpha$.
>
> ---
> **Algorithm 1: RCP1 – Single Sample Robust Conformal Prediction**
>
> - **Input:**
>
> The calibration set
> $\mathcal{D} _ {\mathrm{cal}} = \{(x_i, y_i)\}_{i=1}^n$
>
> The score function
> $s: \mathcal{X} \times \mathcal{Y} \to \mathbb{R}$
>
> the test point
> $x_{n+1}$
>
> target coverage level $1 - \alpha$
>
> perturbation ball $\mathcal{B} _ r$
>
> smoothing scheme $\xi$
>
> - **Output:**
>   $\mathcal{C} _ r(x_{n+1}) \subseteq \mathcal{Y}$
>
> 1. **Step 1.** Draw noise from the smoothing scheme, compute conformity scores for augmented calibration data:
>    ```
>    for i = 1 to n do
>        \epsilon_i ∼ \xi
>        s_i \gets s(x_i + \epsilon_i, y_i)
>    end for
>    ```
>
> 2. **Step 2.**
>    - Compute corrected level:
>      $$1 - \alpha' = c^{\uparrow}[1 - \alpha, \mathcal{B}_r]$$
>    - Compute the $$ (1 - \alpha') $$ quantile with correction:
>      $$q \leftarrow Q(\alpha'; \{s_1, \dots, s_n\})$$
>
>
> 3. **Step 3.** Construct the prediction set for the test input:
>
> $$\mathcal{C} _r (x _{n+1}) = \{ y \in \mathcal{Y} : s(x _{n+1} + \epsilon _{n+1}, y) \ge q \}$$
>
> 4. **Return** $$\mathcal{C} _r(x _{n+1})$$
>
> ---
>
> **Idea behind the proof of Prop 1.**
> We answered this question in the beginning of our response. If the intuition is not yet clear we would be very happy to introduce it again. Thank you for pointing this out. We would definitely add a similar description before the proof of the proposition in the camera ready.
>
> Also we borrowed Zargarbashi and Bojchevski’s definition of $c^\downarrow$ and $c^\uparrow$ without rigorously defining it.  $c^\downarrow[f, x, \mathcal{B}]$ is the certified lower bound for function f in the nearby points around the point x which is $c^\downarrow[f, x, \mathcal{B}] = \lambda \Leftrightarrow \forall x’ \in \mathcal{B}(x): \lambda \le f(x’)$.
>
> We are very thankful for the reviewer’s comment on the readability of the paper. We will surely add descriptions and intuitions to make it more clear. If there is any part that is not yet clear we would be more than happy to further introduce it.

---

> > ### Comment · Reviewer_ABhm · 2025-08-07
> >
> > Thank you for the discussion.
> > I'm trying to re-decypher the proofs.
> >
> > a. In proposition 1, you define q = Q (α; s(Xi + Ei, Yi) : (Xi, Yi) ∈ Dcal}).
> > Here Ei is random (from the smoothing) and Xi,Yi are (discrete) random from Dcal, right?
> > So is the Q function estimated on the discrete mixture of the conditional quantiles for each Xi,Yi?
> >
> > b. c^\downarrow[f, x, \mathcal{B}]$ so that really a lower bound replacement for alpha as alpha', i.e. the minimal probability to aim for after smoothing.
> > [It is really hard that you don't explain these before running into the algorithm].
> >
> > I see that you argue (and it appears in previous papers) that c^downarrow and c^uparrow only depend on the smoothing scheme and ball shape, e.g. line 157 you say that for isotropic Gaussian smoothing on an l2 ball we get a shifted inverted gaussian.
> > I am probably missing, but where do we see the dependence on the function itself (e.g. smoothness, boundedness, etc)?
> >
> > c. If I understand the algorithm, esimtate c^downarrow and c^uparrow are not estimated. Moreover, they are a bound on the complete function or on its worst behavior; is that true?
> > Does this give us conditional coverage or still only marginal coverage in the usual conformal sense?

---

> ### Author Response · Authors · 2025-08-07
>
> We really appreciate the reviewer's reply. We address the questions, and if parts of our response isn't clear, please let us know, and we are more than happy to clarify further.
>
> a. Yes, it is a quantile over the random data point, and its label, augmented with random noise (form the smoothing). In practice the value q is one sample from the distribution of quantile over augmented inputs from Dcal. To know the distribution better, q comes from a distribution made by: (1) sampling a calibration set (2) augmenting inputs x with one random noise from the smoothing scheme. So it is a mixture over the smoothing scheme and the generative data distribution.
>
> b. Sorry again for confusion. $c^\downarrow[f, x, \mathcal{B}]$ itself is the lower bound of function f over any point in B(x). But in our case:
> - It encodes the lower bound of the **smooth** coverage after **perturbation**.
> - We should think of two noises distinctively. One comes from the smoothing scheme, and one is added by the adversary -- we refer to that as perturbation. The smoothing doesn't change the alpha as we calibrate over noise-augmented input, and we also evaluate on noise augmented input (therefore the distribution doesn't change). This lower bound captures the probability of coverage (1 - alpha) when the adversary applies the perturbation.
> - The definition of $c^\downarrow$ is actually for any function f (smooth or non-smooth). In smooth functions (where f has some noise augmentation in it) this lower bound is computed easier, and usually the lower bound is higher -- the function changes slowly (in expectation) when smoothing noise is added to it.
>
> c. Same as the previous papers, for smooth functions we can find this lower (and upper) bound independent to the model (function). In fact the bounds that we use for smooth function (a function evaluated on a smooth augmented input) is computed for the worst case function. So intuitively imagine a model with smoothing in the input space; for computing the bounds we set the function aside and answer that "what is the bound for worst (or any) function and the worst point in B(x) if it is equipped with smoothing and that worst function has same expectation at x". This allows us to treat the model as a black box and apply the same robustness results to any model.
>
> d. In this work yes and in other related works no. In fact the bound only works with exact probability. Therefore if you use it on a variable that its exact mean is not known, then you need to estimate it (with many samples), draw a confidence interval around it, and apply the bound on the edge of that interval.
>
> One novelty of our work is that we find a variable that doesn't need to be estimated -- the 1 - alpha expected coverage is known exactly and therefore the bound can apply without any sampling and confidence intervals.
>
> Regarding the point c. please note that the smoothing-based bound could be computed for any arbitrary smoothing function and threat model B(x), we use L2 and isotropic Gaussian as an example, and in Section 3.1 we discuss how to find the bound for any other smoothing/ball following the results from [22].

---

### Note · Authors · 2025-08-13

We thank the reviewers for reading our paper and for the constructive feedback! We really appreciate that 4/5 reviewers engaged in a fruitful discussion that improved our work. During the discussion period we addressed all concerns, including the following highlights:
- We fixed the flaw in Prop. 1 pointed out by reviewer cNni. We are happy that the reviewer found our response convincing.
- We added an algorithm to address the ambiguity about how RCP1 works in light of the discussion with Reviewers ABhm, and cNni.
- In the discussion with reviewer ABhm we pointed out the reason why prior works require sampling and correction, and why we circumvent it.
- In the discussion with Reviewer MZPH we pointed out the essential difference between our RCP1 and the related work BinCP.

We hope our reply answered the remaining questions from reviewer ABhm. While our definitions are exactly the same as BinCP [24], we agree that our paper would benefit from a more detailed exposition on the background about robustness certificates and the lower/upper bounds. In the camera-ready version we will integrate all of the feedbacks to improve the readability of the paper.

Additionally, we want to highlight that reviewer p6ny did not engage in any discussions with us. To summarize our reply to p6ny:
- The first weaknesses mentioned in their review is wrong. Our robust CP guarantee is for the worst-case (adversarial) perturbation. The guarantee is on average over the data and augmented noise which is different from the perturbation. This type of marginal guarantee is default for most CP methods, including the popular APS (adaptive prediction sets). Averaging over added noise is often not mentioned explicitly, but it's there.
- Our results are independent of the smoothing noise and the threat model. We discuss this in details in Sec 3.1. and later evaluate RCP1 with uniform smoothing against L1 ball in Figure 5-right.
- Novelty comes from two key aspects: orthogonal proofs compared to prior computationally expensive smoothing-based CP, and orders-of-magnitude better guarantees compared to cheaper verifier-based baselines for the same computational budget.

---

### Decision · Program_Chairs · 2025-09-17

**Decision:**

Accept (poster)

**Comment:**

This paper proposes RCP1 (Robust Conformal Prediction with one sample), a method that significantly reduces the computational overhead of robust conformal prediction. Standard RCP methods require many model evaluations per input, whereas RCP1 demonstrates that robustness can be certified with a single randomly perturbed input and a binary certificate. Empirically, RCP1 achieves smaller average prediction set sizes compared to state-of-the-art smoothing-based RCP methods, while reducing cost substantially.

After discussion, all reviewers lean towards acceptance, although the majority still consider that the paper remains borderline.

Strengths

* Novel methodological contribution with clear empirical benefits
* General approach, applicable to any conformity score

Weaknesses

* Writing is dense and presentation/notation is sometimes unclear
* An error was identified in one proof (acknowledged and satisfactorily addressed by the authors)

The authors provided clarifications on the proof, acknowledged typos, and committed to improving presentation. These corrections are expected to address most concerns.